# Neural dynamics of causal inference in the macaque frontoparietal circuit

**Guangyao Qi[1,2†], Wen Fang[1]\*†, Shenghao Li[1,2], Junru Li[1], Liping Wang[1]\***

[1]Institute of Neuroscience, Key Laboratory of Primate Neurobiology, CAS Center for Excellence in Brain Science and Intelligence Technology, Chinese Academy of Sciences, Shanghai, China; [2]University of Chinese Academy of Sciences, Beijing, China

**Abstract** Natural perception relies inherently on inferring causal structure in the environment. However, the neural mechanisms and functional circuits essential for representing and updating the hidden causal structure and corresponding sensory representations during multisensory processing are unknown. To address this, monkeys were trained to infer the probability of a potential common source from visual and proprioceptive signals based on their spatial disparity in a virtual reality system. The proprioceptive drift reported by monkeys demonstrated that they combined previous experience and current multisensory signals to estimate the hidden common source and subsequently updated the causal structure and sensory representation. Single-unit recordings in premotor and parietal cortices revealed that neural activity in the premotor cortex represents the core computation of causal inference, characterizing the estimation and update of the likelihood of integrating multiple sensory inputs at a trial-by-trial level. In response to signals from the premotor cortex, neural activity in the parietal cortex also represents the causal structure and further dynamically updates the sensory representation to maintain consistency with the causal inference structure. Thus, our results indicate how the premotor cortex integrates previous experience and sensory inputs to infer hidden variables and selectively updates sensory representations in the parietal cortex to support behavior. This dynamic loop of frontal-parietal interactions in the causal inference framework may provide the neural mechanism to answer long-standing questions regarding how neural circuits represent hidden structures for body awareness and agency.

**\*For correspondence:**
wenfang@ion.ac.cn (WF);
liping.wang@ion.ac.cn (LW)

†These authors contributed equally to this work

**Competing interest:** The authors declare that no competing interests exist.

## Editor's evaluation

This study investigates the neural basis of the hidden causal structure between visual and proprioceptive signals in the primate premotor and parietal circuit during reaching tasks executed in a virtual reality environment, where information between the two modalities can be dissociated. The key novel result is that premotor neurons represent the integration of bimodal information for small disparities and the segregation for large disparities between the proprioceptive and visual information, while parietal cells show reaching tuning changes that support the updating sensory uncertainty between tasks.

## Introduction

The brain is constantly confronted with a myriad of sensory signals. Natural perception relies inherently on inferring the environment's hidden causal structure (*Deroy et al., 2016*; *French and DeAngelis, 2020*; *Lochmann and Deneve, 2011*). For instance, in the ventriloquism illusion, when the audience is presented with a temporally synchronous but spatially discrepant audiovisual stimulus (e.g., a speech sound from the speaker and a visibly moving mouth of the puppet), they usually infer these audiovisual stimuli are coming from a common source and illusive perceive the speech coming from the

puppet. In the process of building representation of the bodily self, the brain combines, in a near-optimal manner, information from multiple sensory inputs. When a single entity (e.g., the bodily self) evokes correlated noisy signals, our brain combines the information to infer the properties of this entity based on the quality and uncertainty of the sensory stimuli. As a result, behavioral performance often benefits from combining information using uncertainty-based weighting across sensory systems (*Stein and Stanford, 2008*). However, in a natural environment, multiple sensory cues are typically produced by more than one source (e.g., two entities), which should not be integrated in the brain, especially when the superposing cues are sufficiently dissimilar and uncorrelated. Instead, the brain's inferential process of integration fades out, leading to the perception that these cues originate from distinct entities. This process of inferring the causes of sensory inputs for perception is known as causal inference (*Körding et al., 2007*).

Thus far, most of the neurobiological studies of multisensory processing have operated under the assumption that different streams of sensory information can arise from the same source. For example, previous neurophysiological research in monkeys showed that neurons implement reliability-weighted integration on the premise that visual and vestibular signals are from a common source (*Fetsch et al., 2013*; *Morgan et al., 2008*; *Porter et al., 2007*). Therefore, despite the ubiquity of the phenomenon of causal inference and many psychophysical and theoretical research (*Acerbi et al., 2018*; *Dokka et al., 2019*; *Kayser and Shams, 2015*; *Körding et al., 2007*; *Mohl et al., 2020*; *Rohe and Noppeney, 2015*; *Sato et al., 2007*), its neural mechanisms and functional circuits remain largely unknown. Recent neuroimaging studies have started to show the sequential causal inference process in the human brain (*Aller and Noppeney, 2019*; *Cao et al., 2019*; *Rohe et al., 2019*; *Rohe and Noppeney, 2015*; *Rohe and Noppeney, 2016*). However, little was known about the process at the single-neuron level in animals (*Fang et al., 2019*). In particular, the updating of prior and sensory information during causal inference has not been examined in animals.

In the present study, we established an objective and quantitative signature of causal inference at a single-trial level using a reaching task and a virtual reality system in macaque monkeys. We showed that monkeys combined previous experience and current multisensory signals to estimate the hidden common source and, more importantly, subsequently updated both the causal structure and sensory representation during the inference. We then further recorded from the premotor and parietal (area 5 and area 7) cortices of three monkeys to investigate the neural dynamics and functional circuits of causal inference in multisensory processing. Our behavioral and neural results reveal the neural computation that appears to mediate causal inference behavior, including inferring a hidden common source and updating prior and sensory representations at different hierarchies.

## Results

### Behavioral paradigm and hierarchical Bayesian causal inference model

Using a virtual-reality system, we trained three monkeys (monkeys H, N, and S) to reach for a visual target with their nonvisible (proprioceptive) arm while viewing a virtual arm moving in synchrony with a preset spatial visual-proprioceptive (VP) disparity (*Figure 1A*). On each trial of the experiment, the monkeys were required to initiate the trial by placing their hand on the starting position (blue dot) for 1 s and were instructed not to move. After the initiation period, the starting point disappeared, and the visual virtual arm was rotated; this mismatch arm was maintained for 0.5 s as the preparation period. The reaching target was presented as a 'go' signal, and monkeys had to reach toward the visual target within 2.5 s and place their hand in the target area for 0.5 s, referred to as the target-holding period, to receive a reward. Any arm movement during the target-holding period automatically terminated the trial. The proprioceptive drift due to the disparity between visual and proprioceptive inputs was measured at the endpoint of the reach and was defined as the angle difference between the proprioceptive arm and the visual target (the estimated arm) (*Figure 1B*, see details of animal training and reward in Materials and methods). In addition to this VP conflict (VPC) task, two control experiments were conducted: (i) where the visual and proprioceptive signals were perfectly aligned (VP task) and (ii) where there was only a proprioceptive signal (P task). The procedures of the three tasks (VPC, VP, and P) were identical, except that the visual or proprioceptive information presented to monkeys varied according to the context of the experiment (see Materials and methods). Using a block design, the order of three different blocks (tasks) in each training or recording session was randomized.

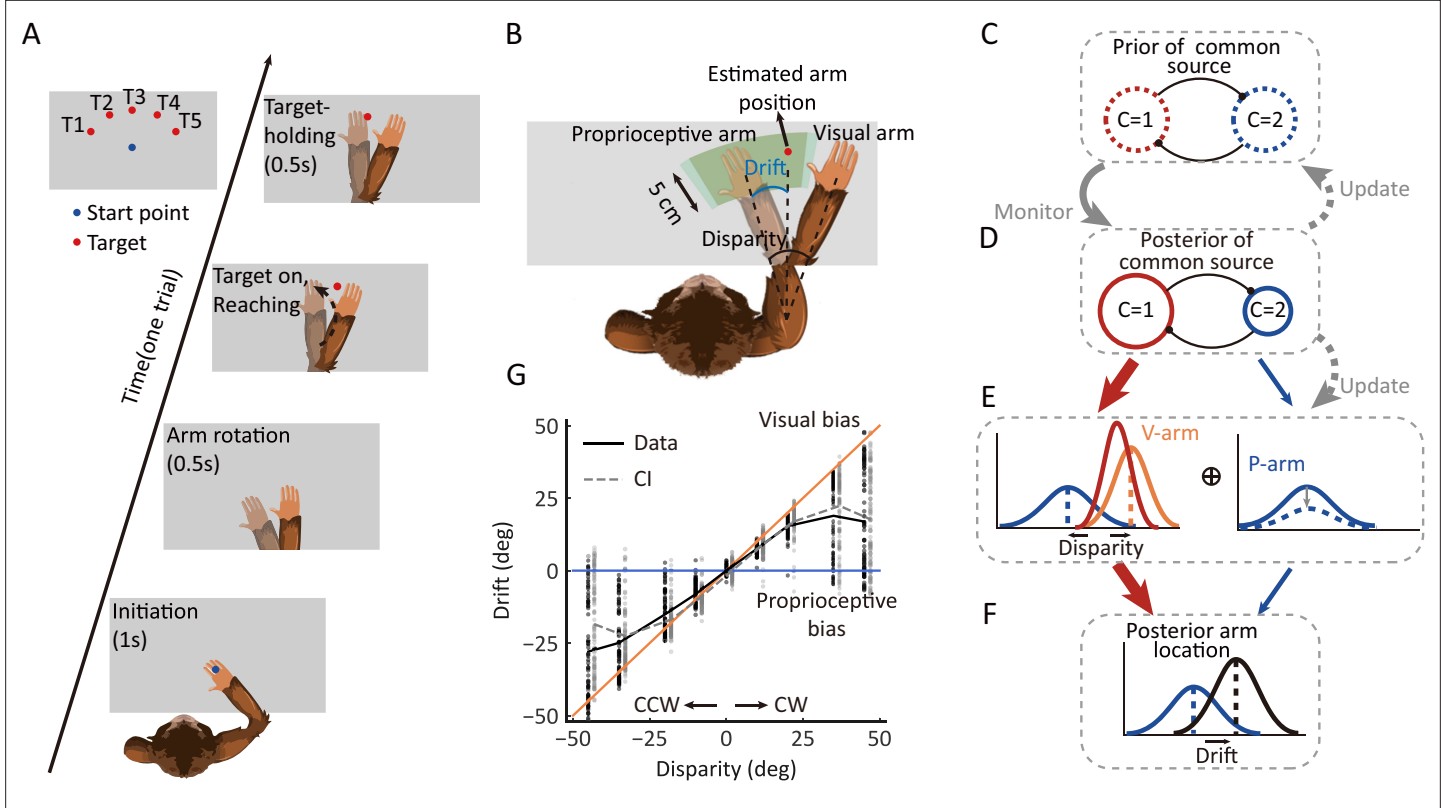

**Figure 1.** Behavioral task, the dynamic hierarchical causal inference model, and proprioceptive drift results. (**A**) Overview of the behavioral task. The monkey was instructed to hold its proprioceptive arm over the starting position (blue dot) to initiate one trial. After the virtual visual arm rotation, a virtual red dot was presented, and the monkey was required to place its proprioceptive arm on the target and hold it to get a reward. (**B**) Schematic drawing of reward area, proprioceptive drift, and the different types of arms (proprioceptive and virtual/visual). Here, proprioceptive drift was defined as the rotated degree from the proprioceptive arm position to the estimated arm position (the same as the target location) measured from the shoulder. The reward area is defined by the green area, which ensures the monkey performed the task rationally and without visual feedback (see animal training in Materials and methods). (**C–F**) Schematic drawing of the dynamic hierarchical causal inference model. V-arm, visual arm signal; P-arm, proprioceptive arm signal; C=1: both V-arm and P-arm come from a common source; C=2: V-arm and P-arm come from different sources. (**G**) Example behavioral results from one session of one monkey (also see *Figure 1—figure supplement 1* and *Figure 1—figure supplement 2*). CCW, counterclockwise; CW, clockwise. The black line represents raw data. The gray line represents the Bayesian causal inference (BCI) model fitting result. The black dots represent experimental trials and the gray dots represent simulated trials by the BCI model.

The online version of this article includes the following source data, source code, and figure supplement(s) for figure 1:

**Source code 1.** Related to *Figure 1G*.

**Source data 1.** Related to *Figure 1G*.

**Figure supplement 1.** Behavior performance and causal inference model predict results in individual monkeys.

**Figure supplement 2.** Histograms of behavior and model-simulated results in an example session.

We used this paradigm to test the hypothesis of the causal inference process, which predicts how the brain infers and updates hidden structures on the basis of multiple sensory inputs.

First, the BCI model encodes probability distributions over the sensory (visual and proprioceptive) signals and incorporates rules that govern how a prior belief about the sensory causal structure is combined with incoming information to judge the event probability in proprioception (*Figure 1C–F*). Thus, the monkey's behavior output (the proprioceptive drift distribution under each disparity) should show the dynamics of integration and segregation, which is the hallmark of causal inference. That is, the drift should increase for small disparities and decrease when the disparities become larger (*Fang et al., 2019*; *Körding et al., 2007*).

Second, according to the model, the prior of common source in the current trial should be modulated by the experience of the environmental structure. Thus, at a single trial level, the prior of a

common source in the current trial should be updated based on the posterior of a common source from previous trials (*Figure 1C and D*).

Third, the sensory uncertainty is also proposed to update to maintain consistency with the prior beliefs of the causal structure of the world (*French and DeAngelis, 2020*). Therefore, the sensory uncertainty should increase when there is a conflict between the proprioceptive and visual signals (e.g., the VPC task) (*Figure 1D and E*).

To test these hypotheses, we adopted the Bayesian causal inference (BCI) model to assess monkeys' behavior and investigated whether the neural activities in multiple brain regions correlate to proposed components in the behavior.

## The probability of common source in monkey's behavior

To examine whether the monkeys inferred the causal structure during multisensory processing, we first examined the proprioceptive drift as a function of disparity in the VPC task. Overall, the three monkeys showed a very consistent behavioral pattern, with the proprioceptive drift increasing for small levels of disparity and plateauing or even decreasing when the disparity became larger (e.g., exceeded 20°) (*Figure 1G*; for data on individual monkeys, see *Figure 1—figure supplement 1*). The BCI model qualitatively explains the nonlinear dependence of drift as a function of disparity. For small disparities, there is a high probability that the proprioceptive and visual signals came from the same source. Hence, the visual information is fully integrated with the proprioceptive information. For large disparities, however, the proprioceptive and visual signals are likely from different sources, leading to a breakdown of integration and consideration of only the proprioceptive information (segregation). In this case, visual information has a weaker weight for perception. Consequently, the effect of disparity on the drift is reduced by shifting integration to segregation. The BCI model quantified the nonlinear dependence between disparity and proprioceptive drift to measure the posterior probability of a common source ($P_{com}$), the consequence of causal inference. We fitted the behavioral data using the BCI model. The results showed two signatures of the $P_{com}$ pattern: (i) the averaged $P_{com}$ decreased as the disparity increased (*Figure 2A*, left) and (ii) within each disparity, especially the large ones, the $P_{com}$ decreased as the proprioceptive drift decreased (*Figure 2A*, right) (see individual monkeys' behavior in *Figure 1—figure supplement 1*).

## $P_{com}$ in the current trial depended on the experience

More importantly, the model posits that not only the inference of the causal structure is based on visual and proprioceptive inputs but also the subsequent updating of (i) the prior belief of causal structure based on the experience (e.g., probability of a common source in the previous trials) and (ii) the uncertainty of sensory signals for the visual and proprioceptive recalibration (*Figure 1C–F*). To test these hypotheses, we first implemented the Markov analysis of the prior belief and $P_{com}$ (see Materials and methods) to see whether the prior probability of a common source ($P_{prior}$) in the current trial depended on the previous $P_{com}$ (*Figure 2B*). The Markov model included the transition probability of $P_{prior}$ between the current ($n$th) and previous ($n^{th}- 1$) trial to account for the trial-by-trial variability in spatial drifts observed in the three monkeys (*Figure 2B*, left). The fit to the model demonstrated that the $P_{com}$ observed in the $n$th trial was significantly affected by that in the previous ($n^{th}- 1$) trial (*Figure 2B*, right, Wilcoxon signed-rank test, p<0.001), indicating that the $P_{com}$ was computed based on both $P_{prior}$ from the previous trial and the sensory inputs, with their disparity, from the current trial. Note that the transition probabilities ($P_{(C=1|C=1)}$ and $P_{(C=1|C=2)}$) remained relatively high (larger than 0.8 in three monkeys) because overall, the number of high $P_{com}$ trials was much more than low $P_{com}$ trials in either training or recording sessions. This was consistent with high baseline $P_{prior}$ in three monkeys (*Supplementary file 1*).

## The common-source belief modulated the sensory uncertainty

We next examined whether the sensory representation is updated to maintain consistency with the causal structure of the environment. That is, the estimates of physical arm locations should tradeoff systematically depending on the current common-source belief (e.g., $P_{com}$ in different tasks: VP, P, and VPC). For example, when the monkey incorrectly infers that the visual and proprioceptive arms come from the same source when a disparity is presented, the uncertainty of proprioception should increase to 'explain away' the conflict between the two inputs. According to this idea, since the block design in

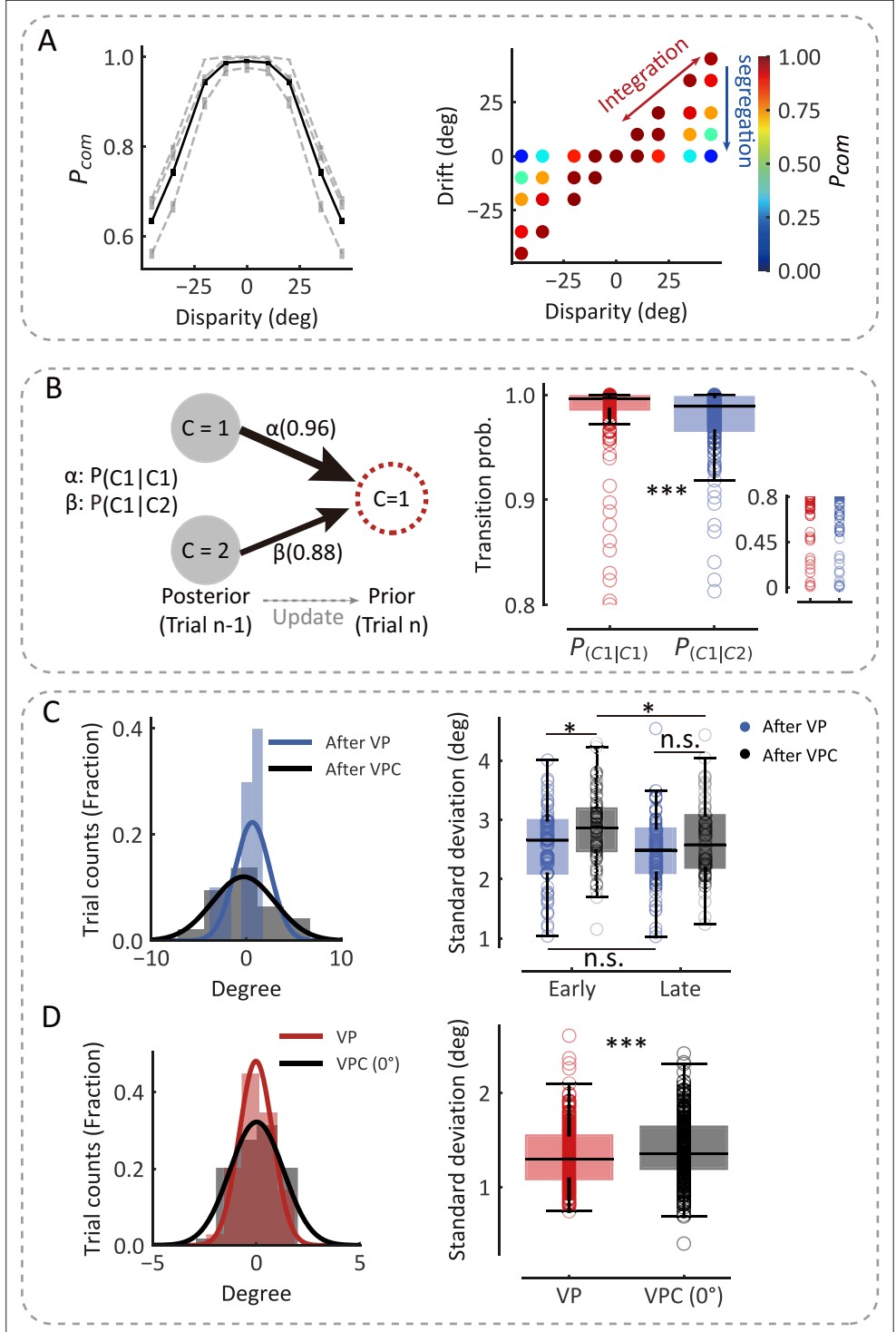

**Figure 2.** The causal inference model predicts the dynamic updating of monkey behavior. (**A**) Left: The average $P_{com}$ as a function of disparity. The black line represents the average $P_{com}$ across monkeys. The dashed lines represent the average $P_{com}$s across sessions of three monkeys separately. Error bars indicate standard errors of the means (SEMs). Right: Model prediction of the $P_{com}$. Each point represents the average $P_{com}$ in each cluster grouped by specific disparity and proprioceptive drift according to the clustering of disparity and drift (see Materials and methods). (**B**) The transition probability from the previous trial's $P_{com}$ to the current trial's $P_{prior}$. Left: The transition probability of an example session. Right: The transition probabilities across all sessions from three monkeys (Wilcoxon signed-rank test, $W=6996.0$, $df = 242$, $p<0.001$, $r_{rb} = 0.52$). Each circle represents a behavior session. The

*Figure 2 continued on next page*

*Figure 2 continued*

most right insert scatter represents a single session's transition probability. (**C**) After-trial effect of sensory updating. Left: The distribution of arm locations in P blocks after visual-proprioceptive (VP) and VP conflict (VPC) tasks in an example session. The solid lines represent fitted Gaussian distributions. Right: The standard deviations of drift in P blocks after VP and VPC tasks across all sessions from three monkeys in early trials (Wilcoxon signed-rank test, $W$=851.0, $df$ = 72, p=0.012, false-discovery rate [FDR] corrected, see Materials and methods, $r_{rb}$ = 0.38) and in latter trials ($W$=1024.0, $df$ = 72, p=0.073, FDR corrected, $r_{rb}$ = 0.24). The uncertainty of P trials after the VPC task in the early part of the session was significantly larger than that in the later part ($W$=917.0, $df$ = 72, p=0.035, FDR corrected, $r_{rb}$ = 0.29); this is not the case for P trials after the VP task ($W$=1086.0, $df$ = 72, p=0.15, FDR corrected, $r_{rb}$ = 0.20). (**D**) Within-trial effect of sensory updating. Left: The distribution of arm locations in VP and VPC (0°) tasks. The solid lines represent fitted Gaussian distributions. Right: The standard deviation of drift in VPC (0°) trials was significantly higher than that in VP trials (Wilcoxon signed-rank test, $W$=10,035.0, $df$ = 237, p<0.001, $r_{rb}$ = 0.29). In (**C**) and (**D**), each circle represents a behavior session. The effect sizes ($r_{rb}$) were performed using the rank-biserial correlation (***Kerby, 2014***). *p<0.05; ***p<0.001; n.s., not significant.

The online version of this article includes the following source data, source code, and figure supplement(s) for figure 2:

**Source code 1.** Related to *Figure 2A-D*.

**Source data 1.** Related to *Figure 2A–D*.

**Figure supplement 1.** Sensory updating is not reflected in the mean of drift.

**Figure supplement 2.** No significant difference of the divergence of eye fixation positions between visual-proprioceptive (VP) and VP conflict (VPC) (0°) tasks (see Materials and Methods, Wilcoxon signed-rank test, $W$=98.0, $df$ = 19, p=0.81).

the current experiment resulted in P trials (in the P task) sometimes following the VPC task and other times following the VP task, we then reasoned that because the overall $P_{com}$ was lower in the VPC task than in the VP task, the uncertainty of proprioception (i.e., the distribution of proprioceptive drifts in the P trials) would be larger after the VPC task than after the VP task. We analyzed the drift variation in P trials and found that, in the early trials (first third of each P block), the uncertainty of P trials following the VPC task was significantly larger than that following the VP task (***Figure 2C***, right, Wilcoxon signed-rank test, p=0.012). The increase in the uncertainty of proprioception was recovered in the late trials (last third of each P block), evident by a significant difference in the uncertainty between early and late P trials (***Figure 2C***, right, Wilcoxon signed-rank test, p=0.035). The decrease in the uncertainty of proprioception was reasonable, as the tradeoff effect in the VPC task gradually recovered.

Furthermore, we hypothesized that if a tradeoff of sensory representation occurs during the process of causal inference, the tradeoff would also affect the uncertainty of VP integration in both VP and VPC tasks. We examined the distribution of proprioceptive drifts using the trials with 0° disparity in the VPC task, in which the V and P information were congruent, and compared it with the distribution in the VP task. As predicted, we found that the variance of the proprioceptive drift was significantly larger in the VPC task than in the VP task (***Figure 2D***, right, Wilcoxon signed-rank test, p<0.001). Note that the difference between VP and VPC (0°) tasks could not be explained by the divergence of eye fixation positions (***Figure 2—figure supplement 2***). As a control, we also investigated whether the mean of drift, representing the perceptual accuracy of the proprioceptive arm, was affected by the causal structure of the environment. We found there was no significant difference between the mean of drift for P trials following the VPC task and that following the VP task in both early parts (***Figure 2—figure supplement 1***, left, Wilcoxon signed-rank test, p=0.37, false-discovery rate [FDR] corrected) and late parts (***Figure 2—figure supplement 1***, right, Wilcoxon signed-rank test, p=0.37, FDR corrected). Besides these, we also found that the mean of proprioceptive drift was not updated in the VPC task compared with the VP task (***Figure 2—figure supplement 1***, right, Wilcoxon signed-rank test, p=0.29). Thus, these results supported the notion of a tradeoff in proprioception according to causal inference environments; that is, sensory representation's uncertainty, not accuracy, is updated dynamically based on the task environment ($P_{com}$).

To summarize the above-described behavioral results, we found that monkeys' proprioceptive drift shows a nonlinear dependency on the disparity between proprioceptive and visual input, which was well explained by the causal inference model. Second, we showed that the $P_{com}$ integrated with VP sensory inputs and is updated by previous experience on a trial-by-trial basis. Third, to maintain a

consistency of causal inference, sensory uncertainty, reflected by the variance of proprioceptive drift, is updated in the inference along with the change of $P_{com}$. Taken together, we established the behavioral paradigm in which monkeys infer the hidden cause by integrating prior information and sensory inputs while dynamically updating both $P_{com}$ and sensory representation. The behavioral responses of the monkeys enabled us to examine the underlying neural mechanisms and functional circuits.

## Causal inference in individual premotor and parietal neurons

Previous studies showed that the premotor and parietal cortices were highly involved in body representation and multisensory perception (see reviews in *Blanke, 2012*; *Graziano and Botvinick, 2002*). In monkeys, bimodal neurons with visual and somatosensory receptive fields were found in both premotor (including F2vr in dorsal premotor and F4/F5 in ventral premotor) and posterior parietal cortices (including area 5 and area 7) (*Fogassi et al., 1999*; *Graziano et al., 2000*; *Graziano and Gross, 1993*; *Graziano and Gross, 1998*; *Graziano et al., 1994*). Specifically, ventral premotor neurons responded to visual stimuli in the space adjacent to the arm (*Graziano and Gross, 1998*; *Graziano et al., 1994*). The bimodal neurons in the parietal cortex (area 5 and area 7) showed to respond to both the real arm position and the seen position of a dummy arm (*Graziano et al., 2000*), which have a significant projection of the premotor cortex (*Graziano and Gross, 1998*). Consistently, human fMRI studies found that the posterior parietal and premotor (dorsal and ventral) cortices selectively respond to visual stimulation near the hand (*Brozzoli et al., 2011*) or the dummy hand near one's corresponding hand (*Blanke et al., 2015*; *Ehrsson et al., 2004*). A human MEG study also revealed that the activities in the prefrontal and intraparietal sulcus were related to the causal inference computation in visual-auditory integration (*Cao et al., 2019*; *Rohe et al., 2019*). Therefore, we determined to record from two brain regions, the premotor cortex (dorsal and ventral, 412 neurons) and parietal cortex (area 5 and area 7; 238 neurons), in the three monkeys performing the reaching tasks (*Figure 3A*, for details, see Materials and methods). We first examined whether neurons in the premotor and parietal cortices during the target-holding period (*Figure 3B*) were selective to basic task components, including condition (VP or P), arm location, and visual disparity. In the premotor cortex, 40% (163/412) of neurons were selective to condition, 23% to arm location, and 37% to visual disparity (*Figure 3—figure supplement 1A*, upper panel). In the parietal cortex, 35% (83/238) of neurons were selective to condition, 27% to arm location, and 31% to visual disparity (*Figure 3—figure supplement 1A*, lower panel, ANOVA, main effect, p<0.05). We also examined the neural representations of the visual and proprioceptive arm locations in each trial during the target-holding period in the VPC, VP, and P tasks, measured by a bias-corrected percent explained variance ($\omega$PEV) (*Figure 3C*). Both brain regions conveyed vital information about the arm location in the three tasks. In the VP and P tasks with no VP disparities, both premotor and parietal cortices showed similar visual and proprioceptive arm information (*Figure 3C*). However, when disparities were introduced in the VPC task, the premotor cortex showed a more robust signal for visual arm information (*Figure 3C*). In contrast, the parietal cortex showed stronger signals for information related to the proprioceptive arm (*Figure 3C*).

Next, to define causal inference response in the VPC task at the single-neuron and single-trial levels, we utilized the VP and P tasks to characterize neural responses, as these tasks involve expected stereotypical behaviors in the two extreme regimes: full integration and segregation. Thus, neurons that are more active during the P task are likely candidates for 'segregation (P) neurons', which exhibited increased activity under the large disparities in the VPC task (*Figure 3D*). By contrast, neurons that are more active during the VP task reflect a preference for integrating congruent VP information and, hence, constitute a natural candidate for 'integration (VP) neurons' (example in *Figure 3—figure supplement 2*). We then implemented a linear probabilistic model which combined how the neural response pattern aligned with the VP and P response profiles and used this model to implement a probabilistic decoding analysis to calculate the probability of VP or P (VP weight = $P_{vp}/[P_{vp} + P_p]$) based on the firing rate in each trial (*Figure 3E*; also see Materials and methods). Thus, a larger VP weight for a single trial denotes a higher probability of integration (high $P_{com}$). We first focused on the target-holding period in a trial, as the neurons could well display their spatial tunings when monkeys holding their arms on the target. We found that both premotor and parietal cortices carry information about $P_{com}$ at the single-neuron (*Figure 3F*; the same example neurons in *Figure 3D*) and population levels (*Figure 3G*; see Materials and methods) during the target-holding period. That is, the VP weight of

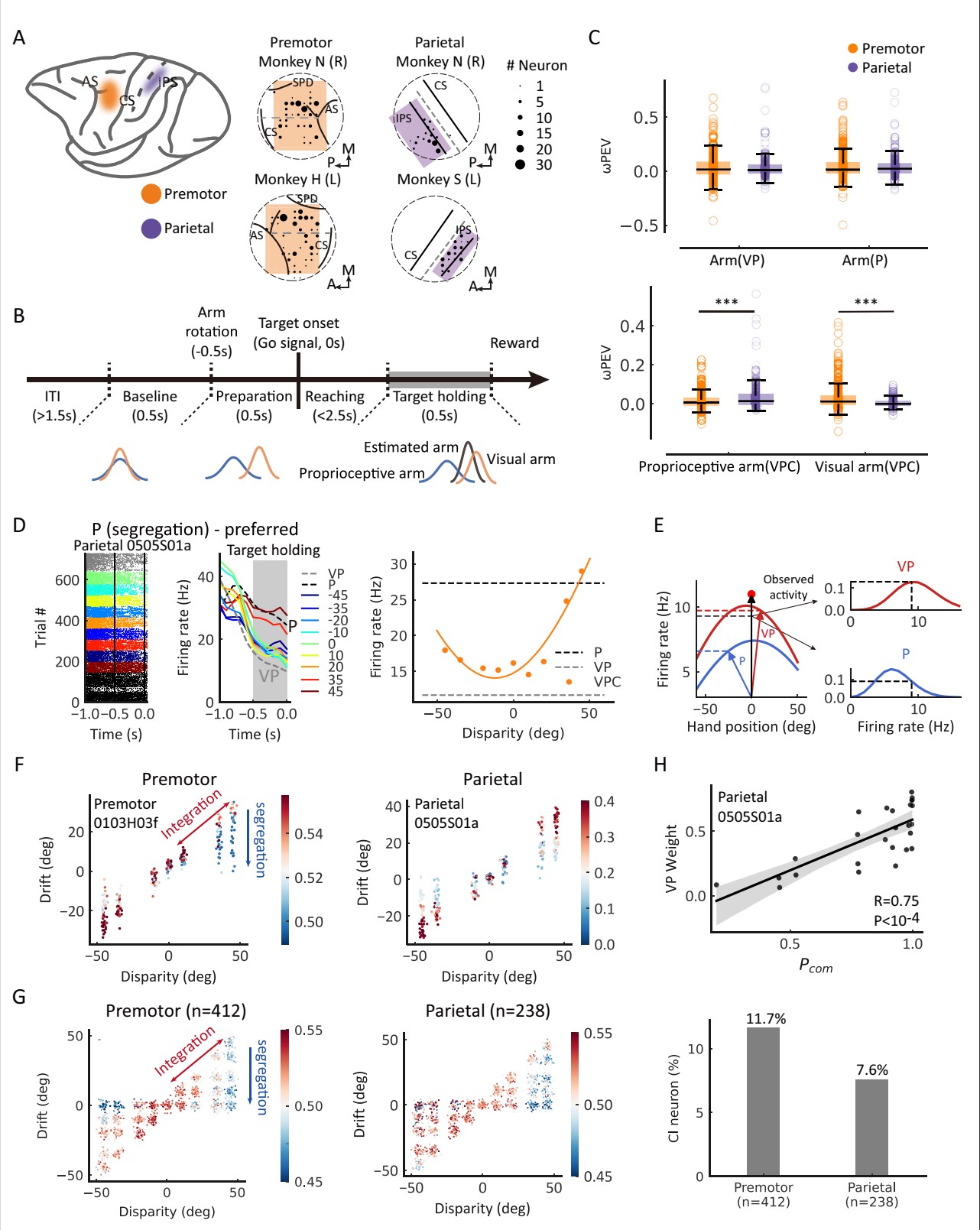

**Figure 3.** Casual inference neurons in premotor and parietal cortices. (**A**) Recording sites. Left: Two regions of interest were recorded through single electrodes in macaque monkeys. Middle and right: Specific recording sites in three monkeys. AS, arcuate sulcus; CS, central sulcus; IPS, intraparietal sulcus; SPD, superior precentral dimple. L, left hemisphere; R, right hemisphere; A, anterior; P, posterior; M, medial. The straight dash gray line separated the dorsal and ventral part of the premotor cortex in the middle panel. The straight dash gray line indicates the middle of IPS and CS. The

*Figure 3 continued on next page*

*Figure 3 continued*

circular dash lines indicate the recording chambers. (**B**) Temporal structure of a single trial for the visual-proprioceptive conflict (VPC) task. (**C**) Neural information of arm locations in premotor and parietal cortices. Upper: No significant difference between the brain regions for the neural information of VP arm (Wilcoxon rank-sum test, $W=0.64$, $df_{premotor} = 411$, $df_{parietal} = 237$, $p=0.52$, false-discovery rate [FDR] corrected, $r_{rb} = 0.030$) and P arm ($W=0.51$, $df_{premotor} = 411$, $df_{parietal} = 237$, $p=0.52$, FDR corrected, $r_{rb} = 0.031$), respectively. Bottom: There were significant differences between the brain regions for both the neural information of proprioceptive arm (Wilcoxon rank-sum test, $W=-3.92$, $df_{premotor} = 411$, $df_{parietal} = 237$, $p<0.001$, FDR corrected, $r_{rb} = 0.18$) and visual arm ($W=6.34$, $df_{premotor} = 411$, $df_{parietal} = 237$, $p<0.001$, FDR corrected, $r_{rb} = 0.30$) in VPC task, respectively. Both brain regions conveyed significant information about the arm location in the three tasks (premotor: VP arm, Wilcoxon signed-rank test, $W = 27,712.0$, $df = 474$, $p<0.001$, FDR corrected, $r_{rb} = 0.35$; P arm, $W = 25,614.0$, $df = 411$, $p<0.001$, FDR corrected, $r_{rb} = 0.40$; proprioceptive arm (VPC), $W=22,316.0$, $df = 411$, $p<0.001$, FDR corrected, $r_{rb} = 0.48$; visual arm (VPC), $W=14,874.0$, $df = 411$, $p<0.001$, FDR corrected, $r_{rb} = 0.65$. Parietal: VP arm, $W=9466.0$, $df = 237$, $p<0.001$, FDR corrected, $r_{rb} = 0.33$; P arm, $W=7414.0$, $df = 237$, $p<0.001$, FDR corrected, $r_{rb} = 0.48$; proprioceptive arm (VPC), $W=3745.0$, $df = 237$, $p<0.001$, FDR corrected, $r_{rb} = 0.74$; visual arm (VPC), $W=10,138.0$, $df = 237$, $p<0.001$, FDR corrected, $r_{rb} = 0.29$). Each circle indicates a neuron. The effect sizes ($r_{rb}$) were performed using the rank-biserial correlation. (**D**) Raster plots and mean firing rates from an example neuron in the parietal cortex that exhibited responses varied with visual disparity, showing the preference for the P task during the target-holding period (gray zones). The yellow curve was fitted with a von Mises distribution. (**E**) Schematic drawing of VP weight analysis (see Materials and methods) in one example trial for the VPC task. In brief, we first mapped the tuning curves of arm position in VP (left red curve) and P (left blue curve) tasks as integration and segregation templates, respectively. Then, during the VPC task, for a single trial, we mapped the visual and proprioceptive arm position onto the these templates to get the probabilities of integration and segregation. Then, we normalized the probability to get the VP weight. (**F**) Two examples of causal inference neurons in premotor and parietal cortices during the target-holding period (the same neurons shown in ***Figure 3—figure supplement 2*** and (**D**), respectively). Each point represents one single trial, and the color represents the value of VP weight. The color bar represents VP weight, larger values indicate higher VP weights (higher probability of integration). (**G**) Population causal inference patterns in two brain regions. Each point was a pseudo-trial that was generated through bootstrapping, and the color represents the value of VP weight. (**H**) An example neuron in the parietal cortex shows the causal inference pattern defined by a significant positive correlation between VP weight and $P_{com}$ (Pearson correlation). Each point represents the average $P_{com}$ and VP weight in a cluster from the behavioral $P_{com}$ pattern. The solid line was fitted with linear regression, and the shaded area indicates the 95% confidence interval. The bar plot represents the fraction of causal inference neurons in the premotor cortex and parietal cortex. ***$p<0.001$.

The online version of this article includes the following source data, source code, and figure supplement(s) for figure 3:

**Source code 1.** Related to ***Figure 3C, D, F, G and H***.

**Source data 1.** Related to ***Figure 3C, D, F, G and H***.

**Figure supplement 1.** Heterogeneity in the responses of neurons to task components in the premotor and parietal cortices.

**Figure supplement 2.** Left and middle: Raster plots and mean firing rates from an example neuron in the premotor cortex that exhibited responses varied with visual disparity that preferred to the P task during the target-holding period (gray zones).

**Figure supplement 3.** Histograms of Pearson correlation coefficients between eye fixation position and visual-proprioceptive (VP) weight.

the neuron or population progressively decreased along with the disparity, and in trials with large disparity (e.g., 35° and 45°), the neuron(s) had a higher VP weight when the drift was large (i.e., the monkey integrated the visual information; thus, a high $P_{com}$ predicted by the BCI model) and shifted gradually toward higher P weights when the drift shifted to 0 (i.e., the monkey segregated the visual information; thus, a low $P_{com}$ predicted by the BCI model). The VP weight was highly correlated with the $P_{com}$ from behavior (***Figure 3H***). Note that the premotor cortex had a slightly higher proportion of causal inference neurons (11.7%) than the parietal cortex (7.6%, Pearson chi-square test, $\chi^2=2.33$, $p=0.063$).

As neuronal activities in the premotor and the parietal cortices are reported to correlate with the eye position in the reaching task (***Buneo and Andersen, 2006***; ***Pesaran et al., 2006***), one might ask whether the $P_{com}$ signals can be explained by the eye position. However, the result showed that the VP weights in the population could not be predicted by eye fixation positions during the target-holding period (***Figure 3—figure supplement 3***).

## Population states encode $P_{com}$ during causal inference

We next focused on the overall populations of neurons in both regions and asked whether and how their population states reflect the uncertainty of causal structure, $P_{com}$. We were guided by the results from single-neuron analyses during the target-holding period described above, in which neurons responsive to high $P_{com}$ (prefer integration) are more likely to show neural tuning similar to that during the VP task, and neurons responsive to low $P_{com}$ (prefer segregation) show a tuning profile similar to that in the P task. We thus hypothesized that neural components or subspaces embedded in the population activity represent the dynamic change in the coding of $P_{com}$ in the VPC task, which would lie between the components representing the VP and P profiles. Furthermore, the computation of $P_{com}$

in the BCI model is determined by the relation and disparities between the visual information from the artificial arm and proprioceptive information from the monkey's actual arm. In other words, according to the model, the causal inference can be constructed before the visual target appears, and the participant uses this information to guide the reach. We thus further hypothesized that the dynamics of the population states also reflect the $P_{com}$ during the preparation period, during which there is no motor planning or preparation.

Thus, we grouped trials from each neuron into high and low $P_{com}$ classes according to the drift under each disparity (high, the top third of the trials [in red]; low, bottom third of the trials [in blue]) (*Figure 4A*). We conducted demixed principal component analysis (dPCA) to visualize any neural component that represents the $P_{com}$ in the VPC task in relation to that in the VP and P tasks (see Materials and methods). dPCA decomposes population activity into a set of dimensions that each explain the variance of one factor of the data (*Kobak et al., 2016*). We included the factors of time, arm location, and $P_{com}$ (*Figure 4B*). In the analysis, VP and P trials were included, which served as the templates of integration and segregation, respectively. As shown in the schema (*Figure 4B*), if the decomposed neural components indeed represent the $P_{com}$, the population activity of high and low classes in this subspace should lie between that of the VP and P classes and the four classes (high, low, VP, and P) should be separated from each other. The dPCA results indicated that the $P_{com}$ components, unrelated to the arm location, represented 29.9% and 20.5% of the total firing rate variance in the premotor and parietal cortices, respectively (*Figure 4C*, in red). Notably, the activity in $P_{com}$ dimensions seems consistent with our hypothesis, demonstrating the dynamics of $P_{com}$ between integration (VP) and segregation (P). In addition, compared to the activity in the parietal cortex, the neural trajectories of the premotor populations showed an earlier divergence in $P_{com}$ dimensions (*Figure 4D*).

To further quantify their dynamics statistically, we trained a linear support vector machine (SVM) using pooled activities in each brain region throughout the entire trial. The dynamic decoding results showed that the $P_{com}$ information is correctly predicted by neuronal population activities in both areas after target onset but is decoded only by premotor neurons during the preparation period when there was no visual target or motor preparation (*Figure 4E*, cluster-based permutation test, p<0.05). Randomization test confirmed the time difference that the $P_{com}$ information occurred significantly earlier in the premotor cortex than the parietal cortex (*Figure 4—figure supplement 3*, randomization test, p<0.01, see Materials and methods). This may suggest that the premotor cortex is where causal inference is computed and sends the information to the parietal cortex during the reaching period.

Next, we tested the relationship between the population activities in the two areas. We performed a joint peri-event canonical correlation (jPECC) analysis, which detects correlations in a 'communication subspace' between two brain regions (*Steinmetz et al., 2019*). In brief, we conducted a canonical correlation analysis for every pair of time points containing the population neural firing rates from the two regions. If the shared neural activity emerges at different times in the two areas, that is, activity in one region potentially leads to activity in the other, then we should observe a temporal offset between them. The jPECC results revealed a significant time lag for activity correlations between premotor and parietal areas in $P_{com}$ dimensions (*Figure 4F*, cluster-based permutation test, p<0.05), suggesting a potential feedback signal of $P_{com}$ from the premotor cortex to the parietal cortex. As a control, we performed the same procedure with misalignment trials (see Materials and methods) to exclude the probability that the observed time lag resulted from the intrinsic temporal property of neuronal activities in these regions. There was no significant time lag between premotor and parietal areas when the trials were misaligned (*Figure 4—figure supplement 1*).

## Experience-dependent $P_{com}$ in the premotor cortex

The behavioral experiments showed that the $P_{com}$ could be updated by previous sensory experience on a trial-by-trial basis. To test the effect of the previous $P_{com}$ on the causal inference in each trial, we examined neural activities during the baseline period in the VPC task before a disparity in the visual and proprioceptive arm was introduced (*Figure 5A*). We again classified the trials according to high and low $P_{com}$. *Figure 5A* depicts the results from an example premotor neuron, showing that during the baseline period, the neural activity exhibited selectivity toward the previous trial's $P_{com}$, and at the same time, its neural trajectories in high and low prior classes lay between the VP and P templates. Of

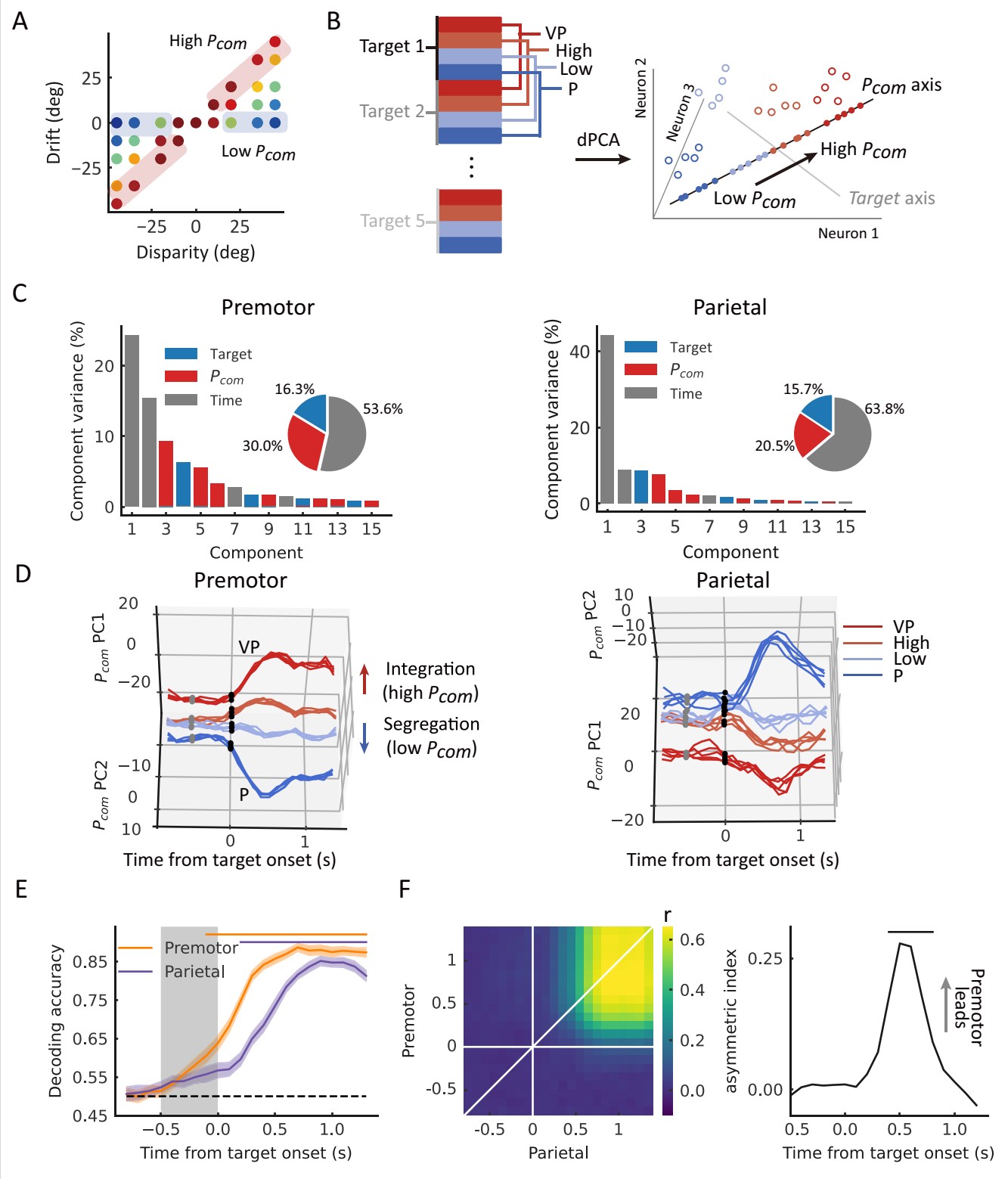

**Figure 4.** Dynamic population decoding of $P_{com}$. (**A**) Schematic drawing of the high $P_{com}$ group (top third of trials) and the low $P_{com}$ group (bottom third of trials) based on the relative drift (drift/disparity). (**B**) Schematic drawing of the demixed principal component analysis (dPCA). All trials of each neuron were grouped into 20 classes (5 targets × 4 conditions, including visual-proprioceptive (VP) and P tasks and high and low groups in the VP conflict [VPC] task). The marginalization matrix was generated by averaging all trials in each class. (**C**) dPCA decomposes population activity into a set of components given the task parameters of interest. (**D**) Temporal evolution of dPCA components of $P_{com}$. The gray points represent the disparity onset; the black

*Figure 4 continued on next page*

*Figure 4 continued*

points represent the target onset. (**E**) Population decoding of $P_{com}$. The decoding accuracy was plotted as a function of time. The gray shaded area represents the preparation period. The horizontal dashed black line represents the chance level. The horizontal solid-colored bars at the top represent the time of significant decoding accuracy (cluster-based permutation test, p<0.05). Shaded areas indicate 95% confidence intervals. (**F**) Joint peri-event canonical correlation (jPECC) results averaged across all sessions. Left: x-axis represents the time of parietal from target onset; y-axis: defines the time of premotor from target onset. The color bar represents the cross-validated correlation coefficient. Right: Lead-lag interactions as a function of time relative to target onset. The horizontal black bar represents the time of significant jPECC asymmetry index versus shuffled data (cluster-based permutation test, p<0.05).

The online version of this article includes the following source data, source code, and figure supplement(s) for figure 4:

**Source code 1.** Related to *Figure 4C–F*.

**Source data 1.** Related to *Figure 4C–F*.

**Figure supplement 1.** Joint peri-event canonical correlation (jPECC) analysis with shuffled temporal alignment trials.

**Figure supplement 2.** Population decoding of $P_{com}$ in the premotor and parietal cortices from Monkey N.

**Figure supplement 3.** The $P_{com}$ information occurred significantly earlier in the premotor cortex than in the parietal cortex.

412 neurons in the premotor cortex, 29 (7.0%) showed such selectivity in the previous trial (*Figure 5—figure supplement 1*).

To further test the relation between baseline neural activity and behavior quantitatively, we examined whether the population activities of these neurons can predict the $P_{com}$ from previous trials. We trained an SVM using pooled activities across recording sessions. The previous $P_{com}$ was only correctly decoded from the baseline activity in the premotor cortex (*Figure 5B*, cluster-based permutation test, p<0.05). Moreover, only recent experience ($n$th−1 trial) had a significant impact on the current trial (*Figure 5C*, permutation test, p<0.001).

As both $P_{prior}$ and $P_{com}$ were represented in premotor neural activities, we wanted to examine their relationship in the neural states. We first found that very few neurons responded to both information types (see *Figure 5—figure supplement 1*). We then hypothesized that $P_{prior}$ and $P_{com}$ might be represented independently at a population level. To validate this hypothesis, we conducted PCA on the population activities during baseline and target-holding periods for $P_{prior}$ and $P_{com}$, respectively. If they are independent, the subspaces of $P_{prior}$ and $P_{com}$ will be near orthogonal, and the PCs of $P_{prior}$ and $P_{com}$ will capture little variance from each other (*Elsayed et al., 2016*). To quantify this, we projected the $P_{prior}$ data onto the $P_{com}$ subspace to calculate the percent variance explained by the $P_{com}$ PCs and repeated the same procedure for the $P_{com}$ data (*Figure 5D*). The results show that the top 10 $P_{prior}$ PCs captured very little $P_{com}$ variance; similarly, the top 10 $P_{com}$ PCs captured very little $P_{prior}$ variance (*Figure 5E*). These results support the hypothesis that the two information types are represented independently in the premotor cortex. However, such independence between $P_{com}$ and $P_{prior}$ could also be caused by their different temporal structures in the task. Thus, we examined their neural dynamics within a trial. *Figure 5F* shows the time course of decoding results of prior and posterior information, where the $P_{prior}$ quickly decreased after the disparity onset. At the same time, the $P_{com}$ information increased and was retained until the end of the trial. These results demonstrated the dynamics in the computation of causal inference, where the information from the last trial is only preserved transiently and then used to integrate with sensory inputs to generate $P_{com}$ information.

## Update sensory uncertainty of arm location in the parietal cortex

Finally, we investigated the neural activities associated with updating sensory uncertainty. The behavior results revealed a significantly greater uncertainty of proprioception in VP trials in the VPC task (low belief of a common source) than in the VP task (high belief of a common source) (*Figure 2D*). We hypothesized that the sensory signals, which were used to make causal inference, in turn, updated their neuronal tunings to match inferred causal structure. We first examined the difference in neural tuning for arm location using the VP trials in the VP and VPC (VPC (0°), trials with no disparity) tasks. To test whether the tuning functions of arm location selective neurons changed between the VP condition and VPC (0°) condition at the single-neuron level, we fitted the tuning curve with the von Mises distribution by using the neuron response in different arm locations (five levels: [−30°, −20°, 0°, 20°, and 30°]) for these two conditions respectively (see Materials and methods). We found that the averaged firing rates during the target-holding period under the VP condition were higher than that

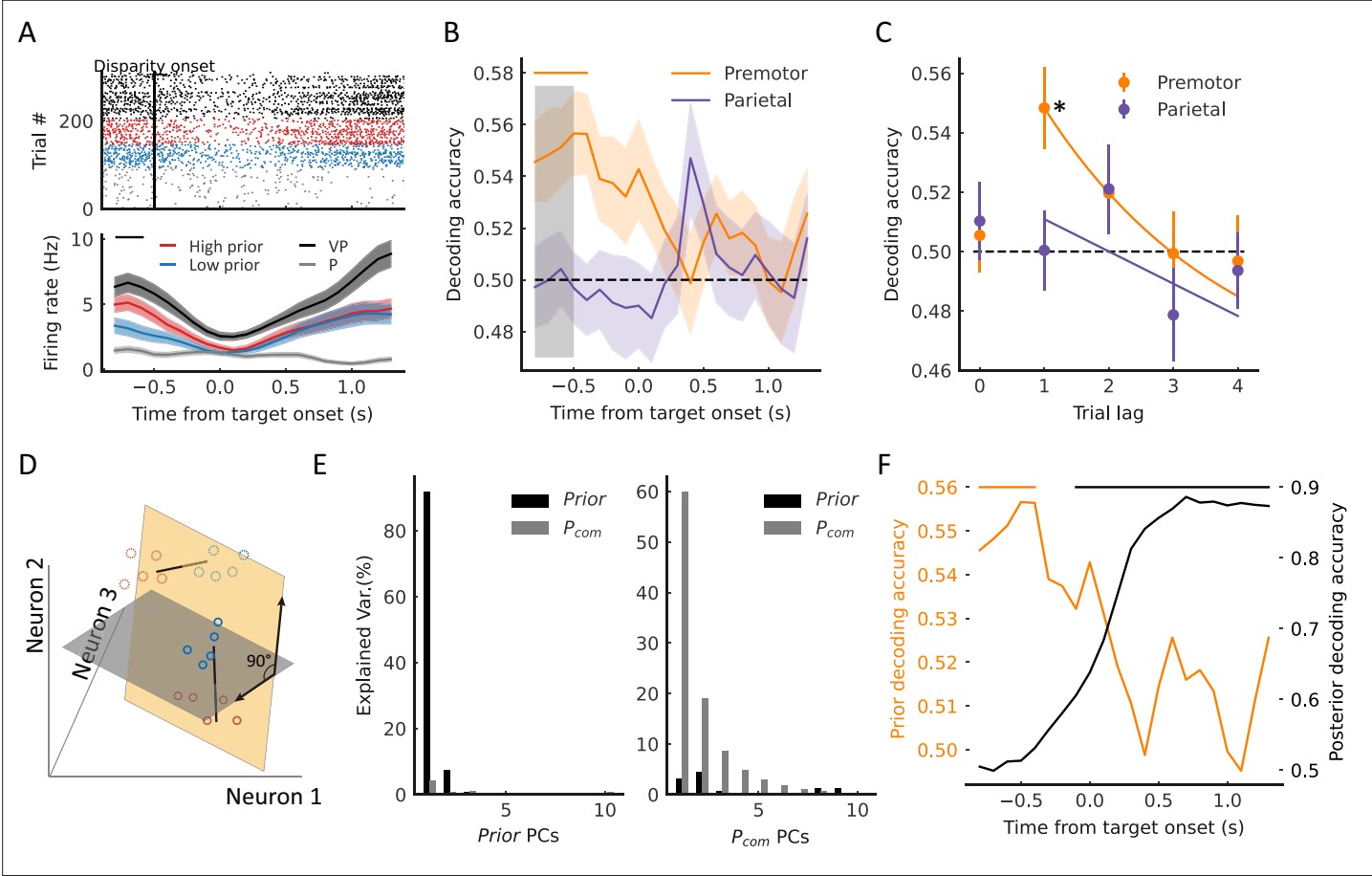

**Figure 5.** Premotor neurons encode prior information (previous trial's $P_{com}$) during the baseline period. (**A**) Example neuron in the premotor cortex showing selectivity to prior information during the baseline period. The trials in the raster plot were sorted by the $P_{com}$ in the previous trial and grouped into high (red dots) and low (blue dots) groups. Bottom: temporal evolution of the average firing rate of 'high prior' and 'low prior' groups. The black horizontal line at the top represents the time window with a significant difference (two-sided $t$-test, $t=2.36$, $p=0.019$). Shaded areas indicate SEMs. (**B**) Dynamic population decoding of prior information ($n$th–1 trial). The gray shaded window represents the baseline period. The horizontal solid colored bar at the top represents the time with significant decoding accuracy with a cluster-based permutation test ($p<0.05$). Shaded areas indicate 95% confidence intervals. The horizontal dashed black line represents the chance level. (**C**) Decoding accuracy of prior trials ($n$th–1 to $n$th–4). Lag 0 represents the decoding of $P_{com}$ in the current ($n$th) trial. The horizontal dashed black line represents the chance level (permutation test, $p<0.001$). The solid lines were fitted with exponential functions. Error bars indicate 95% confidence intervals. (**D**) Schematic drawing of orthogonal subspaces of $P_{prior}$ and $P_{com}$. The solid-line circles represent $P_{com}$ and dotted circles represent $P_{prior}$. Red represents high $P_{com}$, blue represents low $P_{com}$. (**E**) Left: Percentage of baseline-period ($P_{prior}$) data variance (black bars, explained variance: about 99.63%) and target-holding period data variance (gray bars, explained variance: about 8.34%) explained by the top 10 prior PCs. Right: Percentage of baseline-period ($P_{prior}$) data variance (black bars, explained variance: about 11.30%) and target-holding ($P_{com}$) period data variance (gray bars, explained variance: about 99.99%) explained by the top 10 $P_{com}$ PCs. (**F**) Premotor encoded prior information during the baseline period quickly decreased after the disparity onset while the $P_{com}$ information emerged. The orange line represents the population decoding accuracy of $P_{prior}$ ($n$th–1 trial). The black line represents the population decoding accuracy of $P_{com}$. The orange and black horizontal solid-colored bars at the top represent the time with significant decoding accuracy with a cluster-based permutation test ($p<0.05$) for prior information and $P_{com}$ information, respectively. *$p<0.05$.

The online version of this article includes the following source data, source code, and figure supplement(s) for figure 5:

**Source code 1.** Related to *Figure 5A–F*.

**Source data 1.** Related to *Figure 5A–F*.

**Figure supplement 1.** Percentage of prior selective neurons and causal inference (CI) neurons.

**Figure supplement 2.** Population decoding of disparity.

under the VPC condition (0°) in the parietal cortex (*Figure 6—figure supplement 1*, left, Wilcoxon signed-rank test, p=0.017) but not in the premotor cortex (p=0.71). The gain index under VP condition were higher than the VPC condition (0°) in the parietal cortex (*Figure 6—figure supplement 1*, middle, Wilcoxon signed-rank test, p=0.0016, FDR corrected) but not the premotor cortex (p=0.11, FDR corrected). *Figure 6A* (right) shows an example neuron from the parietal cortex tuned to the center (0°) of arm location in the VP task, and the tuning range/uncertainty of the arm location was broader/lower in the VPC task. Here, for visualization purposes, we selected the time point when this neuron demonstrated the highest difference of $\omega$ PEV in the VP trials between VP and VPC tasks for the tuning calculation (*Figure 6A*, left, peak delta $\omega$ PEV). The averaged dynamic spatial selectivity of all neurons revealed a significant decrease of the total spike rate variance explained by the arm location in the parietal cortex but not in the premotor cortex (*Figure 6B*, cluster-based permutation test, p<0.05). Note that the updating of sensory uncertainty was not correlated with the uncertainty of eye position between VP and VPC (0°) tasks (*Figure 2—figure supplement 2*).

Furthermore, at the population level, we performed the SVM decoding analysis of arm locations and found that only the parietal cortex showed a significantly decreased decoding accuracy in the VPC task (*Figure 6C*, cluster-based permutation test, p<0.05). We also confirmed that the change of decoding accuracy in the parietal cortex was significantly larger than the change in the premotor cortex (two-way ANOVA, Condition (*VP and VPC (0°)*)×Region (*premotor and parietal*), significant interaction effect, p<0.05).

## Discussion

Our data of behavior and multi-area neural recordings revealed, for the first time, the dynamic computation of causal inference in the frontal and parietal regions at single-neuron resolution during multi-sensory processing. Complementary to the previous findings focused on the feedforward sequential processing of BCI, the present results demonstrate parallel top-down processing of the hidden variable of $P_{com}$ from the premotor cortex, which monitors the weights of sensory combinations in the parietal cortex. By resolving the experience and causal belief, the hidden causal structure and sensory representation are dynamically updated in the premotor and parietal cortices, respectively.

In the last 15 years, the BCI model has been extended to account for a large number of perceptual and sensorimotor phenomena and a vast behavioral data (*Shams and Beierholm, 2010*). Recent studies have begun to map the algorithms and neural implementation in the human brain. Noninvasive human functional magnetic resonance imaging studies revealed a neural correlation to causal inference in the parietal cortex, and magnetoencephalography showed that frontal neural activities are also involved in the causal inference (*Cao et al., 2019*; *Rohe et al., 2019*; *Rohe and Noppeney, 2015*; *Rohe and Noppeney, 2016*). However, at the single-neuron level, very few studies have examined the neural mechanism in animals. More importantly, none of the human studies have investigated the neural representation of the hidden variable, $P_{com}$. How the frontoparietal circuit contributes to the encoding and updating of $P_{com}$ has not been explored. Our results reconciled and extended previous findings by showing that $P_{com}$ is successively represented by premotor and parietal neural activities (*Cao et al., 2019*; *Fang et al., 2019*; *Rohe et al., 2019*). Unlike previous human imaging studies, which used the final behavioral estimation as the index of the causal inference (*Cao et al., 2019*; *Rohe et al., 2019*), our study directly examined the neural representation and dynamics of the hidden variable $P_{com}$ at single-neuron and neural population levels. We showed that, even within a trial, the inference of a common source was dynamic. We thus propose a dynamic flow of information processing during causal inference, where the $P_{com}$ is estimated from the information of sensory uncertainties and the disparity between them in the premotor cortex and then used for later sensory integration or segregation (model-weighted average) (*Körding et al., 2007*); finally, these signals are maintained in the frontoparietal circuit to guide the reaching behavior (*Archambault et al., 2011*; *Caminiti et al., 2017*; *Cisek and Kalaska, 2005*; *Gail and Andersen, 2006*).

Experience creates our prior beliefs of the surrounding environment. It was proposed that various cognitive functions, such as sensory perception, motor control, and working memory, can be modulated by experience (*Akrami et al., 2018*; *Ernst and Banks, 2002*; *Rao et al., 2012*). Computationally, the prior updating and its modulation of behavior can be well understood within the Bayesian framework (*Badde et al., 2020*; *Beierholm et al., 2020*; *Körding and Wolpert, 2004*; *Rohe et al., 2019*). For instance, by imposing the BCI model in the present study, we showed that prior knowledge of

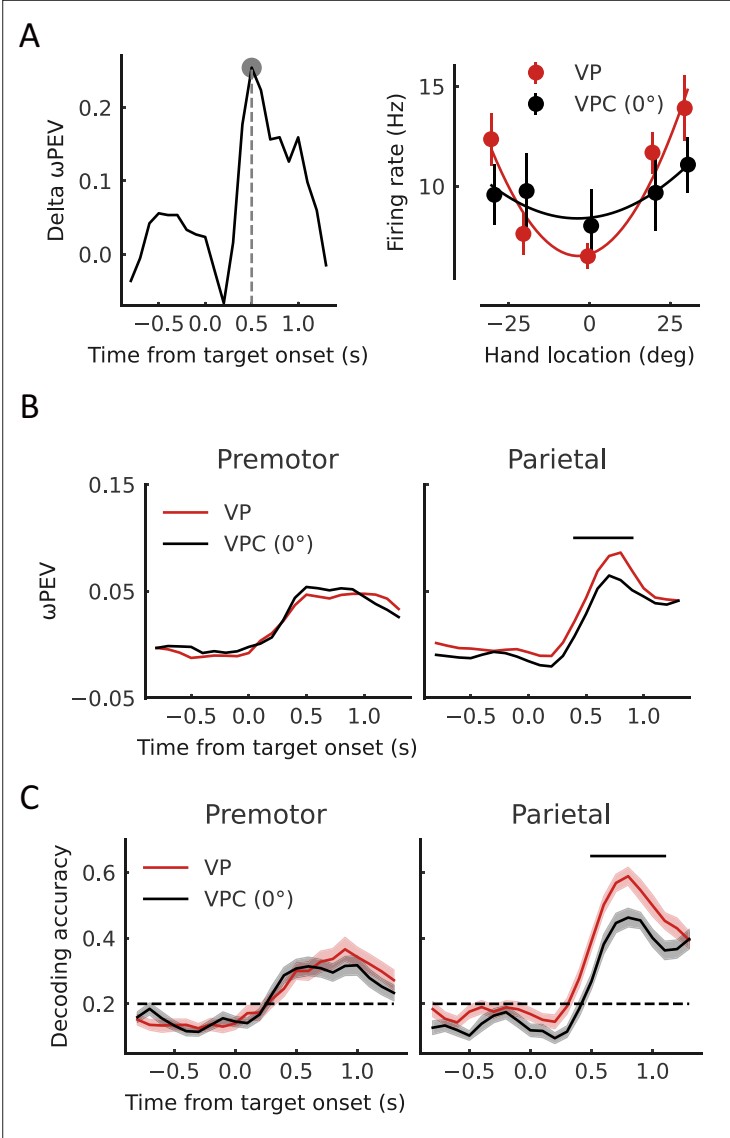

**Figure 6.** Representation of arm location is updated in the parietal cortex. (**A**) Left: The difference of $\omega$PEV between visual-proprioceptive (VP) and VP conflict (VPC) (0°) tasks for an example neuron in the parietal cortex. Right: Snapshot of the arm location tuning for VP and VPC (0°) tasks at the time point showed in the left panel (peak delta $\omega$PEV). At the given time point of this neuron, there is no significant main effect for condition (two-way ANOVA, condition VP and VPC (0°)×hand location; condition, $F_{(1,154)}=1.450$, p=0.23), but for hand location ($F_{(4,154)}=6.736$, p<0.001). The solid curves were fitted with von Mises distributions. (**B**) Dynamic average $\omega$PEV for VP and VPC (0°) tasks. The horizontal bar at the top represents the time bins in which the $\omega$PEV for the VPC (0°) task was significantly lower than that for the VP task (cluster-based permutation test, p<0.05). (**C**) Dynamic population decoding of arm locations. The horizontal bar at the top represents the time bins in which the decoding accuracy for the VPC (0°) task was significantly lower than that for the VP task (cluster-based permutation test, p<0.05). Shaded areas indicate 95% confidence intervals. The horizontal dashed black lines represent the chance level.

The online version of this article includes the following source data, source code, and figure supplement(s) for figure 6:

**Source code 1.** Related to *Figure 6A–C*.

**Source data 1.** Related to *Figure 6A–C*.

**Figure supplement 1.** The comparisons of tuning curve parameters between the visual-proprioceptive (VP) and VP conflict (VPC) (0°) tasks.

a common source is updated by the hidden probability of the common source ($P_{com}$) in the previous trial and then integrated with the sensory inputs in a Bayesian manner. Such prior updating was also reported in a recent sensorimotor study, in which the posterior signals in the frontal cortex were used to update the prior (**Darlington et al., 2018**). Intriguingly, the empirical findings in this study could be reproduced by a biologically plausible recurrent neural network, which suggests that using the feedback of posterior from a Bayesian computation to update prior is an essential feature of a hierarchical recurrent Bayesian model (**Darlington et al., 2018**). From this perspective, the prior updating and its modulation of behavior may also serve as a plausible computational mechanism of multisensory recalibration in various sensorimotor behaviors (**Badde et al., 2020**; **Bruns and Röder, 2015**; **Park and Kayser, 2019**; **Van der Burg et al., 2013**).

The frontoparietal circuit, including the premotor and parietal cortices, has long been recognized as a central area in sensorimotor representations (**Caminiti et al., 2017**; **Caminiti et al., 1991**). Although the present experiments shared many movement features in the reaching task, the key findings of causal inference processing are unlikely to be explained by the kinematical components. First, previous studies have demonstrated that the neuronal activities in the premotor cortex are related to hand kinematics (e.g., hand position, speed, and direction) in the motor planning and execution (**Caminiti et al., 1991**; **Churchland et al., 2006**), which lead the neural activities in the parietal cortex (**Archambault et al., 2011**). However, in our study, the early activities of $P_{com}$ in the premotor cortex cannot be purely induced by the sequential activities of kinematics in the premotor and parietal cortices. Because the $P_{com}$ is abstract information, and its activity pattern is not correlated with any kinematical components. Expressly, under a given value of $P_{com}$, the reaching kinematics can be varied (e.g., the hand position can be anywhere on the table according to the target position and disparity in a given trial). Moreover, the neural signals about $P_{com}$ in the premotor cortex were observed before the target onset, where no motor planning was possible during this period. Thus, our results are consistent with the idea that the high-level information, such as abstract and hidden structures, potential probability of multiple motor options, and VP integration, are encoded in the frontoparietal circuit, which could later integrate with the low-level sensory representations to guide the desired movement (**Cisek and Kalaska, 2005**; **Gail and Andersen, 2006**; **Limanowski and Blankenburg, 2016**).

Second, the dynamic updating of prior and sensory representation proposed a putative mechanism for multisensory recalibration in sensorimotor tasks. At the behavioral level, our results are in accord with the observations that sensory perception is modulated by a multisensory context with sensory conflicts. The BCI theory thus provides a framework to explain how the multisensory context (e.g., the prior of common source) modulates the sensory representations, such as sensory uncertainty in our study and sensory estimation (e.g., spatial localizations) in previous sensorimotor studies (**Badde et al., 2020**; **Bruns and Röder, 2015**; **Park and Kayser, 2019**; **Van der Burg et al., 2013**). The results support the notion of dynamic representations of $P_{com}$ in the present study – the top-down signal of common source from the premotor cortex modulates the spatial tuning in the parietal cortex and then guides hand estimation.

Previous research over the past two decades has revealed that even the perceptions of body ownership and agency are remarkably malleable and involve continuous processing of multisensory information and causal inference (**Kilteni et al., 2015**; **Legaspi and Toyoizumi, 2019**). Thus, our study provides unique data for understanding self-relative awareness (e.g., bodily self-consciousness) in macaque monkeys, showing neural implementation of causal inference at the neural circuit level. Using a VP task, we also identified the hidden components of causal inference in macaque monkeys' parietal and premotor cortices. This is important because, unlike most sensory and cognitive functions, the subjective perceptions of body ownership and agency cannot be directly measured from explicit reports from animals. Using the BCI model and neural activities recorded from multiple brain areas, we can now begin exploring body ownership and agency qualitatively by examining the hidden variable in both behavior and neural representations.

In the BCI framework, there are two key components, inferring the hidden variables (e.g., $P_{com}$) and updating the causal structure and sensory representation. First, our results suggested that the representation and core computation of the hidden common source most likely takes place in the premotor cortex (**Ehrsson and Chancel, 2019**; **Fang et al., 2019**), which is consistent with findings in the body awareness (**Blanke et al., 2015**; **Ehrsson et al., 2004**). Our results were also consistent with previous finding in monkeys that the higher order representations (e.g., the multisensory response of

body recognition) of the body were encoded in both dorsal and ventral premotor cortex and posterior parietal cortex (*Fogassi et al., 1999*; *Graziano et al., 2000*; *Graziano and Gross, 1993*; *Graziano and Gross, 1998*; *Graziano et al., 1994*). Intriguingly, our results seem complementary to previous findings of mirror neuron systems in the premotor and parietal cortices in both humans and monkeys. Typically, a mirror neuron fires both when individual acts and when the individual observes the same action performed by another. That is, the mirror neuron is believed to mediate the understanding of others' behavior (*Jerjian et al., 2020*; *Jiang et al., 2020*; *Pezzulo et al., 2022*). By contrast, the role of causal inference neurons in our study was putatively participating in self-identification and self-other discrimination. Future studies are needed to examine how these two systems work together to identify both self and foreign agents.

Second, the posterior belief of a common source is calculated using a Bayesian approach by integrating prior knowledge and sensory entities, and theoretically, these components should be dynamically updated at different time hierarchies. For example, the prior configuration of the body, known as the body schema in psychology, constrains the possible distribution of the body states but is dynamically updated when the context changes to maintain consistency between the internal body model and sensory inputs (e.g., rubber hand illusion or body illusion) (*Botvinick and Cohen, 1998*; *Kilteni et al., 2015*). Pathological impairment in inferring the sensory source can result in somato-paraphrenia, in which the patient declares that their body part belongs to another person despite the visual and proprioceptive signals from the common source of their body (*Keromnes et al., 2019*). Similarly, schizophrenia patients suffering from delusions of the agency have shown impairments in updating their internal causal structures. They show a deficit in detecting the source of their thoughts and actions and thus incorrectly attribute them to external agents (*Haggard, 2017*). Therefore, although we demonstrated the neural representations and their updating by using the multisensory and reaching task in monkeys, the computational mechanism and underlying neural circuits might contribute to learning and inference in any task that relies on causal inference.

## Materials and methods
### Experimental model and subject details
All animal procedures were approved by the Animal Care Committee of the Center for Excellence in Brain Science and Intelligence Technology, Institute of Neuroscience, Chinese Academy of Sciences (Permit Number: CEBSIT-2020034), and were described previously in detail (*Fang et al., 2019*). Three male adult rhesus monkeys (*Macaca mulatta*; Monkeys H, N, and S, weighting 6–10 kg) participated in the experiment. During the experiment, the monkeys were seated comfortably in the monkey chairs, and their heads were fixed. All monkeys were implanted with chambers for recordings.

### Method details
Some of the following methods are similar to those previously published (*Fang et al., 2019*).

### Apparatus
The monkeys were seated in front of a chest-height table on which a lab-made virtual reality system was placed (*Fang et al., 2019*). During the experiment, the monkey's left arm (and the right arm in the case of Monkey H, who was right-handed) was placed in the system and blocked from sight. A CCD camera (MV-VEM120SC; Microvision Co., China) captured the image of the monkey's arm reflected in a 45° mirror. This image was projected to the rear screen by a high-resolution projector (BenQ MX602, China). Therefore, when the monkey looked in the horizontal mirror suspended between the screen and the table, the visual arm image appeared to be its real arm on the table. The lower edge of the screen was aligned to the table edge. The monkey's trunk was close to the edge of the table, and the left shoulder was aligned with the midline of the screen. Using the OpenCV graphics libraries in C++ (Visual Studio 2010; Microsoft Co., Redmond, WA, USA), the arms image and the visual target were generated and manipulated. Using CinePlex Behavioral Research Systems (Plexon Inc, Dallas, TX, USA), sampled at 80 Hz, the hand position was tracked and recorded. The tracking color marker was painted onto the monkey's first segment of the middle finger, which was not visible after adjusting the light exposure settings of the video.

## Behavioral task procedures

The monkey was trained to report its proprioceptive arm location by reaching for a target in a VPC causal inference task (*Figure 1A*; *Fang et al., 2019*). The monkey initiated a trial by placing its hand on the starting point (a blue dot with a 1.5 cm diameter) for 1000 ms and was instructed not to move. After the initiation period, the starting point disappeared, and the visual arm was rotated (within one video frame, 16.7 ms) for the VPC task. The rotation was maintained for 500 ms (the preparation period). After that, the reaching target was presented as a 'go' signal. The monkey had to reach the target (chosen from T1 to T5 randomly trial by trial [*Figure 1A*]) within 2500 ms and hold its hand in the target area (see as follows) for 500 ms to receive a drop of juice as the reward. Any arm movement during the target-holding period automatically terminated the trial. The rotated arm was maintained throughout the entire trial along with the arm movement. The intertrial interval (ITI) was ~1.5–2 s, after which the monkey was allowed to start the subsequent trial. During the ITI, the visual scene was blank. Under the VPC task, across trials, the visual arm was randomly presented with a disparity of 0°, ±10°,± 20°, ±35°, or ±45° (+, clockwise [CW]; −, counterclockwise [CCW] direction) from the subject's proprioceptive arm, with its shoulder as the center point. The starting point was fixed 25 cm away from the monkey's shoulder. The target position was selected randomly trial by trial from one of five possible positions located on an arc (a ±4° jitter was added to the original position trial by trial to ensure the monkey did not perform the task by memorizing all the target positions).

Besides the VPC task, the monkey was also instructed to perform a VP congruent and P task during the recording session. The only difference between the VPC and VP task was that during the entire trial under the VP task, the visual arm was always congruent with the proprioceptive arm. The only difference between VP and P tasks was that during the single trial for the P task, the visual arm information was blocked starting from the onset of the preparation period.

Each VPC block contained 55 trials in which the nine disparities and five targets were randomly combined. Each VP and P block had 27 trials in which five targets randomly occurred in every single trial. In one recording session, typically, one or two P blocks were given first to ensure that the monkey performed the task with its proprioceptive arm, and then in the following blocks, VP, P, and VPC tasks were randomly mixed. One recording session contained more than three VP and P blocks and more than eight VPC blocks.

## Target (with reward) area

To ensure the monkeys indeed performed the reaching-to-target task with their proprioceptive hand, under the VPC task, the reaching target area (with reward) was defined as follows: the radial distance from the hand to the center of the target was less than 5 cm to ensure that the monkey did reach out to the target; with the target as the center, the azimuth range was set from [−7 (8 for some sessions, same below) + rotation degree/disparity] to +7° when the rotation degree was negative (counterclockwise), and from –7° to [+7 + rotation degree/disparity] when the rotation degree was positive (clockwise). As shown in *Figure 1B* (green zone), the reward area ensured the monkey performed the task rationally and without visual feedback. That is monkey's reaching position between two extreme conditions: one is that the monkey reaches the target purely relying on the visible arm (the drift is equal to the disparity); the other is that the monkey relies on the proprioceptive arm (the drift is equal to zero). Only the correct trials (when the monkey's arm was located within the reward zone) were used in the subsequent analysis.

## Electrophysiology

Extracellular single-unit recordings were performed as described previously (*Fang et al., 2019*; *Merchant et al., 2013*) from three hemispheres in three monkeys. Briefly, under strictly sterile tasks and general anesthesia with isoflurane, a cylindrical recording chamber (Crist Instrument Co., Inc, Hagerstown, MD, USA) of 22 mm diameter was implanted in the premotor cortex and the parietal cortex (area 5 and area 7). We collected the structural magnetic resonance images (MRI) of three monkeys (3T, Center for Excellence in Brain Science and Intelligence Technology, Institute of Neuroscience, Chinese Academy of Sciences), while they were in an MRI-compatible Horsley-Clarke stereotaxic apparatus. The location of the recording chamber on each animal was determined by the atlas with the origin at the Ear Bar Zero (*Saleem and Logothetis, 2012*). The centers of implanting recording chambers were [right: 20.0 mm; forward: 10.0 mm] for the premotor cortex in Monkey N, [left: 21.9 mm;

forward: 24.9 mm] for the premotor cortex in Monkey H, [right: 14.7 mm; forward: 1.1 mm] for the parietal cortex in Monkey N, and, [left: 17.0 mm; forward: 3.5 mm] for the parietal cortex in Monkey S. During the recording session, glass-coated tungsten electrodes (1–2 MΩ; Alpha Omega, Israel) were inserted into the cortex via a guide tube using a multi-electrode driver (NAN electrode system; Plexon Inc, Dallas, TX, USA). All isolated neurons were recorded regardless of their activity during the task, with the recording locations varying from session to session. At each location, the raw extracellular membrane potential was sampled at 40 kHz. On-line raw neural signals were processed offline to obtain a single unit by Offline Sorter (Plexon Inc, Dallas, TX, USA). All spike data were re-sorted using off-line spike sorting clustering algorithms (Offline Sorter, PCA) (**Merchant et al., 2013**). With manual adjustments, only well-isolated units were considered for further analysis (signal-to-noise is larger than 3). The sorted files were then exported in MATLAB format for further analysis in MATLAB (Mathworks, Natick, MA, USA) and Python (The Python Software Foundation).

## Quantification and statistical analysis

All statistical analyses were implemented with scripts written in MATLAB or Python. In the premotor cortex, 412 neurons were recorded from two monkeys (231 neurons from Monkey H and 181 neurons from Monkey N); in the parietal cortex (area 5 and area 7), 238 neurons were recorded from two monkeys (116 neurons from Monkey N and 122 neurons from Monkey S). As all monkeys' behavior and model fitting results were similar, for all analyses, data were combined across monkeys. All related statistics are reported in the figure legends.

## Analysis of behavior data
### BCI model

To capture the uncertainty of causal structure, the core of causal inference, the BCI model described in a previous study (**Fang et al., 2019**) was adopted. In the present study, the BCI framework included three models: (i) the full-segregation model, which assumes that visual and proprioceptive estimates of the arm's locations are drawn independently from different sources ($C=2$) and processed independently; (ii) the forced-fusion model, which assumes that visual and proprioceptive estimates of the arm's locations are drawn from a common source ($C=1$) and integrated optimally, weighted by their reliabilities; and (iii) the BCI model, which computes the final proprioceptive estimate by averaging the spatial estimates under full-segregation and forced-fusion assumptions weighted by the posterior probabilities of a common source. Here, the BCI model assumes that both visual and proprioceptive location information ($S_V$ and $S_P$) are represented as $x_V$ and $x_P$ in the neural system, respectively, which are drawn from the normal distribution with sensory noise [$N(S_V, \sigma_V)$, $N(S_P, \sigma_P)$]. The causal inference structure is determined by the joint distribution of two sensory signals (sensory likelihood) and the prior probability of a common source ($P_{prior}$). Thus, according to the Bayesian rule, the posterior probability of a common source ($P_{com}$) is calculated as follows:

$$p(C = 1|x_V, x_P) = \frac{p(x_V, x_P|C = 1) P_{prior}}{p(x_V, x_P|C = 1) P_{prior} + p(x_V, x_P|C = 2)(1 - P_{prior})}$$

and the two sources of probability are $p(C = 2|x_V, x_P) = 1 - p(C = 1|x_V, x_P)$. Here, the likelihood of observed data ($x_V, x_P$) given common source [$p(x_V, x_P|C = 1)$] is calculated as follows **Körding et al., 2007**:

$$p(x_V, x_P|C = 1) = \frac{exp\left[\frac{-1}{2} \frac{(x_V - x_P)^2 \sigma_{Pr}^2 + (x_V - \mu_{Pr})^2 \sigma_P^2 + (x_P - \mu_{Pr})^2 \sigma_V^2}{\sigma_V^2 \sigma_P^2 + \sigma_V^2 \sigma_{Pr}^2 + \sigma_V^2 \sigma_P^2}\right]}{2\pi \sqrt{\sigma_V^2 \sigma_P^2 + \sigma_V^2 \sigma_{Pr}^2 + \sigma_P^2 \sigma_{Pr}^2}}$$

where $N(\mu_{Pr}, \sigma_{Pr})$ represents a prior distribution of arm locations. In this experiment, the $\mu_{Pr}$ was set to 0 and $\sigma_{Pr}$ was set to 10,000 to approximate a uniform distribution.

If the system completely 'believes' the two sensory signals are from different sources (full-segregation situation), the proprioceptive arm position is estimated independently from the visual information, as follows:

$$\hat{S}_{P,C=2} = \frac{\frac{x_P}{\sigma_P^2} + \frac{\mu_{Pr}}{\sigma_{Pr}^2}}{\frac{1}{\sigma_P^2} + \frac{1}{\sigma_{Pr}^2}}$$

If the system completely 'believes' there is only a common source for the two sensory signals (forced-fusion situation), then the estimate of arm position is determined by the optimal integration rule, as follows:

$$\hat{S}_{VP,C=1} = \frac{\frac{x_V}{\sigma_V^2} + \frac{x_P}{\sigma_P^2} + \frac{\mu_{Pr}}{\sigma_{Pr}^2}}{\frac{1}{\sigma_V^2} + \frac{1}{\sigma_P^2} + \frac{1}{\sigma_{Pr}^2}}$$

Here, we used the model average decision function to estimate final arm location (*Fang et al., 2019*):

$$\hat{S}_P = p\left(C = 1|x_V, x_P\right)\hat{S}_{VP,C=1} + \left(1 - p\left(C = 1|x_V, x_P\right)\hat{S}_{P,C=2}\right)$$

In the model simulation, the proprioceptive arm position at the end of the trial was set to zero ($S_P = 0$), so that the visual arm position is the VP ($S_V = disparity$). In the task, monkeys were required to report their proprioceptive arm position; thus, only the proprioceptive estimate was simulated.

## Model fitting

To estimate the best-fitting model parameters in the BCI model, for each recording session, an optimization search was implemented that maximized the log-likelihood of each model given the monkey's data under the VPC task. The prior probability of a common source ($P_{prior}$) and visual and proprioceptive standard deviations, $\sigma_V$ and $\sigma_P$, respectively, were set as free parameters to be optimized. For each optimization step, 5000 trials per disparity were simulated to obtain the distribution, and the sum log-likelihood of the observations given the model was calculated for each disparity. Then, the parameters were optimized by minimizing the sum log-likelihood using a genetic algorithm (ga function in MATLAB). The procedure was the same as for the optimal integration model, except that there were no causal structures and only two free parameters ($\sigma_V$ and $\sigma_P$) needed to be optimized. All simulation and optimization processes were performed in MATLAB. Only correct trials were included.

## Model comparison

To determine the model that best explained the data at the group level using the Bayesian information criterion (BIC), a Bayesian random-effects model comparison was used (*Rigoux et al., 2014*). $BIC = -2LL + k \times ln\left(n\right)$, where $LL$ denotes the log-likelihood, $k$ is the number of free parameters, $n$ is the total number of data points, and $ln$ is the natural logarithm. The BIC is a criterion for model selection among a finite set of models; models with lower BIC are generally preferred. Finally, the better model was identified at the group level by the exceedance of the probability based on all sessions of monkeys' BICs (*Wozny et al., 2010*). We used the exceedance probability to evaluate how likely it is that any given model is more frequent than all other models in the comparison set.

The models' goodness-of-fit was reported using the coefficient of determination ($R^2$) (*Fang et al., 2019*),

$$R^2 = 1 - exp\left[\frac{-2}{n}\left\{LL\left(\hat{\beta}\right) - LL\left(0\right)\right\}\right]$$

where $LL(\hat{\beta})$ and $LL(0)$ denote the log-likelihoods of the fitted and the null model, respectively, and $n$ is the number of observations. The null model assumes that monkeys report the perceived arm position randomly over the disparity range from the leftmost to the rightmost. Thus, a uniform distribution over this span was predicted.

## *P*prior updating in causal inference

To evaluate how the previous posterior probability of a common source ($P_{com}$) influences the prior probability of a common source ($P_{prior}$), a Markov process was adopted to model the updating of $P_{prior}$. That is,

$$p^n_{(C=1)} = p_{(C=1|C=1)} * p^{n-1}_{(C=1|Data)} + p_{(C=1|C=2)} * \left(1 - p^{n-1}_{(C=1|Data)}\right)$$

where $p_{(C=1)}$ and $p_{(C=1|Data)}$ denote $P_{prior}$ and $P_{com}$ respectively, and $n$ denotes the $n$th trial under the VPC task. Two prior states were included: $C=1$ (a common source) and $C=2$ (two different sources) at each trial. $p_{(C=1|C=1)}$ denotes the transition probability from a common source ($C=1$) to a common source ($C=1$), and $p_{(C=1|C=2)}$ denotes the transition probability from different sources ($C=2$) to a common source ($C=1$). For statistical significance analysis between $p_{(C=1|C=1)}$ and $p_{(C=1|C=2)}$, the Wilcoxon signed-rank test was used for paired data.

Note both $P_{prior}$ and $P_{com}$ are latent variables. During the model fitting, we first used the BCI model (as mentioned before) to search the overall $P_{prior}$, $\sigma_P$, and $\sigma_V$ for each session/day, which were used as initial parameters in the subsequent Markov model. The $\sigma_P$ and $\sigma_V$ were fixed during the Markov model fitting. For all subsequent trials (except the first trial), both $P_{prior}$ and $P_{com}$ are unknown. As time goes on, starting from the first trial, the $P_{com}$ of the current trial is obtained through the BCI model, and the $P_{prior}$ of the next trial is obtained through the integration probability ($P_{com}$) or separation probability ($1 - P_{com}$) which are multiplied and added by the corresponding transition probability. Here, we fitted the observed data-drift to get the two free parameters transition probability. Through the transition probability, we define the influence of the $P_{com}$ of the previous trial on the $P_{prior}$ of the next trial.

## Updating of proprioceptive representation

To evaluate whether the primary sensory representation was modulated by the belief of causal structure, the proprioceptive variance within and after VPC tasks was compared to the baseline condition. For the within effect, the proprioceptive drift was calculated using the trials with 0° disparity in the VPC task and trials in the VP task (baseline condition). Here, the standard deviation (SD) of proprioceptive drift was used as a measurement for the uncertainty of proprioceptive representation, in which higher SD indicates higher uncertainty and vice versa. The mean of the proprioceptive drift for each target was normalized to zero. For the after-effect, the SDs of proprioceptive drift under the P task were compared between after the VP task and after the VPC task. To characterize the temporal dynamic of the proprioceptive updating (after-effect), trials in the first third and trials in the last third of the P task were compared. As a control, a similar analysis was conducted for the raw mean of proprioceptive drift (*Figure 2—figure supplement 1*). For statistical significance analysis, Wilcoxon signed-rank test was used for paired data.

## Eye movement analysis

We trained the monkeys to perform the task without their eye fixed, but the eye movement during the recording sessions was recorded. To examine whether the updating of sensory uncertainty was correlated with the uncertainty of eye position between VP and VPC (0°) tasks. We identified the eye fixation position at the target-holding period. We examined the divergence of eye fixation position in VP and VPC (0°) tasks (see below). The average distance from the central point was used to measure the divergence at each target for each session. Each session's divergence was obtained by averaging all the trials (see follows). The eye-tracking data were imported into MATLAB using EDF Converter (SR Research). The fixations and saccades in eye movements were separated with the default algorithm of the software with the velocity (30°/s), acceleration (8000°/s$^2$), and motion thresholds (0.1°), respectively. The fixation positions were averaged during the target-holding period of each trial.

The normalized divergence of 2D eye fixation positions at each target was determined as follows:

$$Divergence = \frac{1}{n} \sum_{i=1}^{n} \|z_i - c\|$$

where $n$ is the sample size, $c = (c_1, c_2)$ is center of the eye fixation position, and $z_i = \{x_i, y_i\}$ is $i$th eye fixation position. The divergence was averaged across different target positions for each session. $\|z_i - c\|$ indicates the Euclidean distance between $c$ and $z_i$. For statistical significance analysis, Wilcoxon signed-rank test was used for paired data.

Moreover, to examine whether the neural activity of $P_{com}$ was correlated with the eye position, we calculated the Pearson correlation coefficients between eye fixation position and VP weight. We found

that there was no correlation between the VP weight and the eye fixation position at the population level for both regions (the premotor and parietal cortices) at both horizontal and vertical directions (*Figure 3—figure supplement 3*, Wilcoxon signed-rank test, premotor (horizontal): p=0.11; premotor (vertical): p=0.86; parietal (horizontal): p=0.35; parietal (vertical): p=0.87). Note that the recorded eye movement data used in this analysis included 78 sessions for the premotor cortex and 45 sessions for the parietal cortex.

## Correction for FDR

In all cases, we used the Benjamini-Hochberg procedure (*Benjamini and Hochberg, 1995*) to control FDR at an $\alpha$=0.05 level, as follows. The p-values of a given set of hypothesis tests were sorted in ascending order, {$p_1$, $p_2$, …, $p_n$}, and we found the first rank $i_\alpha$ such that $p_{i_\alpha} \leq i_\alpha \times 0.05/n$ . Then we considered tests to be significantly above chance (rejecting null hypotheses) for all $p < p_{i_\alpha}$ .

## Preprocessing of single-unit data

To estimate continuous time-dependent firing rates, timestamps of spiking events were resampled at 1 kHz and converted into binary spikes for single trials. Spike trains were then convolved with a symmetric Hann kernel (MATLAB, MathWorks),

$$convolvedw\left(n\right) = A\left(1 - cos\left(2\pi \frac{n}{N}\right)\right), 0 \leq n \leq N\left(N = L - 1\right)$$

where $A$ is a normalization factor ensuring the sum of the kernel values equals 1. Window width $L$ was set to 300 ms. Single neurons were included in the analysis only if they had been recorded for a full set of tasks (VP, P, and VPC tasks with nine disparities: 0°, ±10°, ±20°, ±35°, and ±45°).

Peri-stimulus time histograms (PSTHs) were then calculated for four periods of interest in a trial: (i) the baseline period (500 ms before the onset of visual arm rotation), (ii) the preparation period (500 ms after the onset of the visual arm rotation), (iii) the target-onset period (1000 ms after the onset of target onset), and (iv) the target-holding period (500 ms after the onset of target-holding). To smooth the firing rate at each time point, the neural firing rate was calculated by averaging in sliding windows (window size, 400 ms; step size, 100 ms) in a single trial (*Fried et al., 2011*; *Gu et al., 2016*), resulting in 22 time bins of mean firing rate for every single trial for subsequent dynamic analysis.

## Task selective neurons

To examine whether neurons in the premotor and parietal cortices during the target-holding period were selective to basic task components, including condition (VP or P task), arm location, and visual disparity. For each neuron, we conducted a two-way ANOVA in two datasets. One dataset contains the VP and P tasks (condition (two levels: VP and P tasks)×arm location (five levels: [–30°, –20°, 0°, 20°, and 30°]); the response variable is the mean firing rate during the holding period of each neuron). If a main effect of condition (or arm location) in the two-way ANOVA was found (p<0.05), this neuron was classified as a condition (or arm location) selective neuron. The other dataset is the VPC task (the visual disparity (nine levels: [–45°, –35°, –20°, –10°, 0°, 10°, 20°, 35°, 45°])×target position (five levels: [–30°, –20°, 0°, 20°, and 30°]); the response variable is the mean firing rate during the holding period of each neuron). If the main effect of visual disparity in the two-way ANOVA was found (p<0.05), this neuron was classified as a visual disparity selective neuron.

## Tuning curve analysis of arm location selective neurons

To investigate whether the tuning functions of arm location selective neurons (*Figure 3—figure supplement 1*) changed between VP condition and VPC (0°) condition at single-neuron level, we fitted the tuning curve with a reduced von Mises function by using the neuron response in different arm location (five levels: [–30°, –20°, 0°, 20°, and 30°]) for these two conditions separately. Here, VPC (0°) condition represents the trials in VPC condition where the disparity equals to 0. And the fitting function was defined as:

$$fr\left(x\right) = b + a * cos\left(x - \mu\right)$$

where $b$ is the spontaneous firing rate of the neuron, $a$ is defined as the gain index, and $\mu$ is preferred arm location. $fr\left(x\right)$ represents the firing rate when the arm location is $x$. We analyzed the

spontaneous firing rate of the neuron, gain index, and preferred arm location between VP condition and VPC (0°) condition in both premotor and parietal cortices (***Figure 6—figure supplement 1***).

## Causal inference neuron

To measure the representation of a single neuron for causal inference on a single trial, the probability that a single neuron would integrate or segregate the sensory information on a single trial was calculated (***Fang et al., 2019***). The basic assumption here is that in a single trial under the VPC task, if the neuron is more inclined to represent integrated information, then its firing rate will be closer to its response under VP tasks and farther away from the response under P tasks, and vice versa. The normalized weight of integration (VP weight) was calculated as follows:

(1) First, obtain the neuron response to the arm position under P and VP tasks and fit the von Mises distribution to get the tuning curve.

(2) Under VPC tasks, obtain the current visual arm and the real arm positions, and at the same time, obtain the neuron's firing rate when the arm is in the corresponding position under VP and P tasks, $\lambda_{VP}$ , and $\lambda_P$, respectively.

(3) The VP and P templates can be generated through the Poisson distribution:

$$Pr_{VP}\left(X = k\right) = \frac{\lambda_{VP}^k e^{-\lambda_{VP}}}{k!}$$

$$Pr_P\left(X = k\right) = \frac{\lambda_P^k e^{-\lambda_P}}{k!}$$

(4) According to the corresponding probabilities, $Pr_{VP}$ and $Pr_P$ in the two templates are obtained, and the integration weights for this neuron in the VPC task can be obtained through standardization:

$$VPweight = \frac{Pr_{VP}}{\left(Pr_{VP} + Pr_P\right)}$$

To quantitatively describe whether a single neuron is encoding causal inference, the correlation between $P_{com}$ and VP weight is calculated. The logic is as follows: the $P_{com}$ can be used to measure the degree of integration or segregation of sensory information at the behavioral level, whereas VP weight can measure this characteristic at the electrophysiological level. Therefore, if a neuron is performing causal inference, there should be a significant positive correlation between the $P_{com}$ and VP weight for the corresponding behavior. Neurons that (i) respond to VP/P tasks and (ii) for which $P_{com}$ and VP weight are significantly positively correlated in the final holding stage are called causal inference neurons. The specific algorithm was as follows:

1. First, obtain neurons with significant selectivity under VP and P tasks (condition selective neuron, see Materials and methods: Task selective neurons).
2. According to proprioception drift, all trials were divided into 29 classes. Continuous drift values were grouped into nine clusters: < −35°, [−35° −25°], [−25° −15°], [−15° −6°], [−6°+6°], [+6°+15°], [+15°+25°], [+25°+35°], >+35°. To be noticed, ±6° covers approximately 99% of drift distribution under the VP and P task. Thus, for the disparity of 0°, there was only one cluster [−6°+6°]. Since the distribution of drift becomes wider (higher variance) the larger the disparity, the more clusters would be assigned for the big disparity. For example, for the disparity ±45°, there were five clusters of drifts. $P_{com}$ and VP weight were assigned for each class by averaging all trials within it. The Pearson correlation coefficient was then calculated between $P_{com}$ and VP weight. If the $P_{com}$ and VP weight were correlated significantly and positively (p<0.05 and $r$>0), the neuron was called a causal inference neuron.

## Population pattern of causal inference

To visualize the VP weight pattern at the brain region level, the VP weight of each trial of a single neuron under VPC tasks was calculated and then divided into 29 clusters as described above. Then, the bootstrap method was used to randomly select 50 trials from each cluster for averaging. This was repeated 50 times to obtain the VP weight (50×29) of a neuron for visualization. This results in a 50 × 29 × N matrix, where N indicates the number of neurons in each brain region (all neurons were used). The trial corresponding to each neuron was averaged to obtain a 50×29 matrix. The VP weights of a brain region were visualized in a heatmap.

## High/low $P_{com}$ groups

To characterize the dynamic representation of the $P_{com}$ in the entire session, all trials in a recording session were divided into high $P_{com}$ trials and low $P_{com}$ trials based on the relative proprioception drift (RD). Each trial's relative proprioception drift (RD = drift/disparity) was calculated. The basic idea was that the larger the $P_{com}$, the more likely the monkey would integrate the visual and proprioceptive information, and the corresponding RD is closer to 1. The top third and bottom third of the trials were designated the high $P_{com}$ class and the low $P_{com}$ class, respectively. These grouping methods were verified by the dPCA.

## dPCA

The method for dPCA was adopted from that published in a previous study (*Kobak et al., 2016*). Time, target position/arm location (−30°, −20°, 0°, 20°, and 30°), and $P_{com}$ (VP, P, high $P_{com}$, and low $P_{com}$) were combined to obtain the marginalized covariance matrix of the three. The neurons whose trial number was not less than five under a single condition were selected for dPCA. Population activity was then projected on the decoding axes and ordered by their explained total variance for each marginalization.

## Information encoded by individual neurons

The percentage of explained variance (PEV) (*Buschman et al., 2011*) was used to measure the basic task components encoded by a single neuron, in which PEV reflected the degree to which the variance of a single neuron can be explained for a specific task component. Generally, PEV can be expressed as a statistical value of $\eta^2$ , that is, the variance ratio between groups to the total variance. As the statistical value of $\eta^2$ has a strong positive bias for a small sample, the unbiased $\omega^2$ statistical value ($\omega$ PEV) (*Olejnik and Algina, 2003*) was used.

To evaluate the information about the locations of the proprioceptive arm, visual arm, and estimated arm encoded by a single neuron in the VPC task, an analysis of covariance was used to decompose the variance, and the $\omega$ PEV was calculated. In detail, for a single neuron, $\omega$ PEV was calculated for each type of arm when setting the other two types of arm locations as covariates. The whole reaching space was divided into 11 parts from −45° to 45° to transform it from a continuous variable to a discrete variable. A nonparametric Wilcoxon rank-sum test was used for unpaired data for statistical significance analysis comparing two brain regions.

The $\omega$ PEV was calculated in each time bin to characterize the temporal dynamics of neural information under VP and VPC (0°) tasks. The baseline was defined as the period 500 ms before the onset of visual arm rotation. A one-sided, paired Wilcoxon signed-rank with FDR correction determined the time bins significantly different from the baseline. The time bins showing significant differences between VP and VPC (0°) tasks were determined by a cluster-based permutation test (*Gramfort et al., 2013*).

## Population decoding analysis

### Decoding of $P$com

The population decoding analysis of $P_{com}$ was performed by the linear SVM classifiers with the scikit-learn toolbox (*Pedregosa et al., 2011*). All neurons were included in this analysis without considering their $P_{com}$ selectivity. The classifier was trained to classify the $P_{com}$ (high/low $P_{com}$) with neural activity (PSTHs) from each brain region. All recording sessions were pooled to form a pseudo-population. Neurons with more than 50 trials in each $P_{com}$ group were included in this analysis. Tenfold cross-validation was then implemented by splitting the neural data into 10 subsamples, each randomly drawn from the entire dataset. Decoders were then trained on nine of the subsamples and tested on the remaining one. This process was repeated 10 times to obtain the decoding accuracy by averaging across all 10 decoders. This cross-validation process was repeated 1000 times, and the overall decoding accuracy was taken as the mean across the 1000 repetitions. The decoding analysis was conducted for all time points. The significance of decoding accuracy was determined by comparing the mean decoding accuracy to the null distribution from the shuffled data. The significant time duration was determined using a cluster-based permutation test for multiple comparisons across time intervals (permutations = 5000; cluster-level statistic: sum of the $t$ values in a cluster; auxiliary cluster

defining threshold $t$=3) (*Gramfort et al., 2013*). For visualization, we plotted the mean of decoding accuracy with 95% confidence interval using 50 repetitions.

To test whether the premotor cortex neurons encode $P_{com}$ earlier than parietal cortex, a randomization test was performed between them. Neurons with more than 50 trials in each $P_{com}$ group were included in this analysis. The corresponding numbers (here, 200 neurons per region) of neurons were randomly exchanged between the paired regions 1000 times to generate a null distribution (chance level) of time lags, and the significance was determined by a permutation test of the true time lag from the original data and the null distribution (*Panichello and Buschman, 2021*).

### Decoding of $P$prior

Neurons with more than 50 trials in each $P_{com}$ group (high and low $P_{com}$ groups, same as for the $P_{com}$ decoding analysis described above) were selected for the $P_{prior}$ updating decoding. The decoding procedure was the same as described for '*Decoding of $P_{com}$*' unless the trials were sorted and labeled by the previous trial's $P_{com}$ ($n^{th}-1$ to $n^{th}-4$) under the VPC task. The statistical significance was determined by a cluster-based permutation test (*Gramfort et al., 2013*).

### Subspace overlap analysis

PCA was performed on neural activities during the baseline period and during the target-holding period. The first 10 principal components (PCs) during each period were used to obtain the $P_{prior}$ and $P_{com}$ subspaces. To test the overlap of these subspaces, the baseline-period activity was projected onto the $P_{prior}$ subspace, and the percent variance explained relative to the total variance of the baseline period data was quantified; similarly, the target-holding period activity was projected onto the $P_{com}$ subspace, and the percent variance explained relative to the total variance of the target-holding period data was quantified (*Elsayed et al., 2016*).

### Decoding of arm locations

All arm locations were separated into five spatial bins: −30°, −20°, 0°, 20°, and 30°. The basic decoding procedure was the same as described above for '*Decoding of $P_{com}$*'. Neurons with more than six trials in each arm location bin were selected. Leave-one-out cross-validation was then implemented, and this process was repeated 1000 times to obtain the averaged decoding accuracy. The decoding analysis was conducted for all time points. Statistical significance for decoding accuracy was determined by comparing the mean decoding accuracy to the null distribution from shuffled data. The time bins with significant differences between tasks (VP and VPC (0°)) were determined by the cluster-based permutation test for multiple comparisons across time intervals (*Gramfort et al., 2013*).

## jPECC analysis

To test the relationship between population activities in the two brain regions, the jPECC method described in a previous study (*Steinmetz et al., 2019*) was utilized. First, the neuronal responses in two brain regions under the same behavior conditions, namely, high $P_{com}$ and low $P_{com}$, were aligned. Then, a PCA was conducted across time and trials to reduce the dimensionality to obtain the first 10 PCs for each brain region. The trials were then divided into 10 equal parts (training set and testing set) for cross-validation (10-fold cross-validation). The PCs of the training set of each brain region were used to perform canonical correlation analysis to obtain the first pair of canonical correlation components (L2 regularization, $\lambda$=0.5). Then, the PCs of the testing set from each brain region were projected onto the first pair of canonical correlation components, and the correlation was determined by the Pearson correlation coefficient between these projections from each region. This analysis was performed for each pair of time bins to construct a cross-validated correlation coefficient matrix. Fifty trials for each group (high $P_{com}$ and low $P_{com}$) from each brain region were randomly selected by bootstrapping in this analysis. Finally, a heatmap was obtained by averaging the correlation coefficient matrix repeated 1000 times.

To quantify the lead-lag relationship of information exchange between brain regions, an asymmetric index was calculated by diagonally slicing the jPECC matrix from +300 ms to +300 ms relative to each time point (*Steinmetz et al., 2019*). For time point $t$, the average correlation coefficient across the left half of this slice (i.e., the average along a vector from $[t-300, t+300]$ to $[t, t]$) was subtracted from the right half of this slice (from $[t, t]$ to $[t+300, t-300]$) to yield the asymmetry index. To test the

leading significant time point across brain regions, the data from neurons in these brain regions were exchanged, and the above-described analysis was repeated 1000 times to obtain the null distribution of the asymmetric index. Then, a cluster-based permutation test was performed to test whether the symmetric index was significantly greater than the chance level (*Gramfort et al., 2013*).

To further exclude the possibility that the observed lead-lag relationship resulted from the intrinsic properties of neuronal activities rather than the encoded information in these regions, all trials in each brain region were shuffled to ensure that the inter-region trials were not aligned. Then, the analysis was repeated as described above to obtain the asymmetric index.

Note that, due to the limitations of the asynchronous recording (the premotor and parietal neurons were grouped from different individual animals and only Monkey N was recorded in both areas), further studies are required to clarify the dynamics and functional interactions between regions using a simultaneous recording.

## Resource availability

### Lead contact
Further information and requests for resources should be directed to and will be fulfilled by the Lead Contact, Liping Wang (liping.wang@ion.ac.cn).

### Materials availability
This study did not generate new unique reagents.

## Acknowledgements

We thank Florent Meyniel and Tianming Yang for their comments on the manuscript, and Xinjian Jiang, Jian Jiang, and Juntao Feng for experimental assistance. This work was supported by the National Science and Technology Innovation 2030 Major Program 2021ZD0204204 to WF, the Shanghai Municipal Science and Technology Major Project 2021SHZDZX to LW, the Lingang Laboratory Grant LG202105-02-01 to LW, the Strategic Priority Research Programs XDB32070201, the Strategic Priority Research Programs XDB32070201 to LW, and the National Natural Science Foundation of China 32100830 to WF.

## Additional information

### Funding

| Funder | Grant reference number | Author |
|---|---|---|
| National Science and Technology Innovation 2030 Major Program | 2021ZD0204204 | Wen Fang |
| Shanghai Municipal Science and Technology Major Project | 2021SHZDZX | Liping Wang |
| Lingang Laboratory Grant | LG202105-02-01 | Liping Wang |
| Strategic Priority Research Programs | XDB32070201 | Liping Wang |
| National Natural Science Foundation of China | 32100830 | Wen Fang |

The funders had no role in study design, data collection and interpretation, or the decision to submit the work for publication.

### Author contributions
Guangyao Qi, Conceptualization, Data curation, Software, Formal analysis, Validation, Investigation, Visualization, Methodology, Writing – original draft, Writing – review and editing; Wen Fang, Conceptualization, Data curation, Formal analysis, Validation, Investigation, Visualization, Methodology,

Writing – original draft, Writing – review and editing, Funding acquisition; Shenghao Li, Data curation; Junru Li, Methodology; Liping Wang, Conceptualization, Supervision, Funding acquisition, Investigation, Writing – original draft, Project administration, Writing – review and editing

### Author ORCIDs
Guangyao Qi http://orcid.org/0000-0003-0479-7320
Wen Fang http://orcid.org/0000-0001-7748-6772
Liping Wang http://orcid.org/0000-0003-2038-0234

### Ethics
All animal procedures were approved by the Animal Care Committee of Center for Excellence in Brain Science and Intelligence Technology, Institute of Neuroscience, Chinese Academy of Sciences (Permit Number: CEBSIT-2020034).

### Decision letter and Author response
Decision letter https://doi.org/10.7554/eLife.76145.sa1
Author response https://doi.org/10.7554/eLife.76145.sa2

## Additional files

### Supplementary files
• Transparent reporting form
• Supplementary file 1. Model parameters and fitting evaluations of four models for monkeys.

### Data availability
Source data files have been provided for Figures 1–6. Code and dataset have been uploaded to Dryad (https://doi.org/10.5061/dryad.rr4xgxd9h).

The following dataset was generated:

| Author(s) | Year | Dataset title | Dataset URL | Database and Identifier |
| --- | --- | --- | --- | --- |
| Qi G, Fang W, Li S, Li J, Wang L | 2022 | Code and dataset for neural dynamics of causal inference in the macaque frontoparietal circuit | https://doi.org/10.5061/dryad.rr4xgxd9h | Dryad Digital Repository, 10.5061/dryad.rr4xgxd9h |

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
