## [Editor Report]

This study investigates the neural basis of the hidden causal structure between visual and proprioceptive signals in the primate premotor and parietal circuit during reaching tasks executed in a virtual reality environment, where information between the two modalities can be dissociated. The key novel result is that premotor neurons represent the integration of bimodal information for small disparities and the segregation for large disparities between the proprioceptive and visual information, while parietal cells show reaching tuning changes that support the updating sensory uncertainty between tasks.

---

## [Decision Letter]

**Decision letter after peer review:**

Thank you for submitting your article "Neural dynamics of causal inference in the macaque frontoparietal circuit" for consideration by *eLife*. Your article has been reviewed by 3 peer reviewers, including Hugo Merchant as the Reviewing Editor and Reviewer #1, and the evaluation has been overseen by Tirin Moore as the Senior Editor. The following individuals involved in review of your submission have agreed to reveal their identity: Hugo Merchant (Reviewer #1).

Essential revisions:

The present results are novel and provide important notions. However, we have a series of important concerns.

1) First, the paper is difficult to follow. The authors should use a simpler and more intuitive framing of the paper to make it more accessible to a general audience. A reader needs to integrate a large part of the results to really have an idea of what is the paper about and how the authors tested their hypothesis. Indeed, the writing in part uses words or phrases that are 'unusual' or imprecise, which makes the text difficult to understand.

2) Another important concern is that the paper does not show the basic response properties of neurons on both areas across tasks. It is not clear how the neural activity changes between the VP, P, and VPC tasks. It is a change in preferred direction, in the width of the tuning function or on the gain modulation? We strongly suggest providing two 'layers' of information, one based on standards plotting and analysis, to be followed by a second layer of data investigation, addressed to more detailed computational aspects.

3) Statistics:

Results of model fitting: The methods mention that the authors compared three models: segregation, fusion and causal inference. However, in the results (Table S1) only two models are presented. The results for segregation should be reported for the sake of completeness. The Ppriori is high, e.g. 0.98 and 0.999. This makes me wonder in how far the R2 and the EPs' can be so much different between causal inference and fusion. Considering e.g. model averaging decision function for BIC, a prior of 0.999 would make fusion and BCI nearly identical. I feel that either the Methods section is lacking important details about the models (See above) or there is either a mistake in the analysis or the reported numbers in the table. The analysis of Bayesian models seems to lack major details, the statistical reporting is below standard (missing effect sizes, degrees of freedom, lack of individual data in figures), the study shows many unjustified parameter choices and key results seem to lack statistical support: not all statements about differences between parietal and premotor cortex seem supported by a direct statistical comparison. Further, while three monkeys contributed data, for only one does the study report data from both brain regions; this makes the claim of a difference between brain regions rather weak and this shortcoming needs to be clearly acknowledged. The actual underlying data (e.g. how single neuron responses are converted to tuning curves; how decoding accuracies vary across neurons) is not shown, which makes it difficult to interpret the robustness of the results. In particular, as the units of analyses vary tremendously between Figures (experimental blocks, neurons, pseudo-epochs, etc).

The updating of the believe about the sensory causal structure is a central component of this work. The authors present this as well established aspect of the BCI model (l. 170ff). However, most previous studies used a static model that was fit to the aggregate data of an entire experiment without taking the trial history into account (as in the cited Koerding et al. 2007 paper). AS some recent work has incorporated such trial dependencies, it would be important to acknowledge these studies and to explain the novelty of the present work (e.g. Rohe et al. NatComm 2019; Beierholm et al. *eLife* 2020; Badde et al. Cognition 2020 may also be relevant).

Analysis of recalibration: The analysis of P trials after the VP or VPC task effectively looks at what is known as trial-wise multisensory recalibration (see e.g. Bruns et al. Scientific Reports 2015; Park and Kayser *eLife* 2019; van der Burg et al. J Neurosci 2013; Wozny et al. JNeurosci 2011; Badde et al. Cognition 2020; there is extensive literature on this both in the spatial and the temporal domain!). It seems awkward to investigate this recalibration of uniusensory judgements without alluding to previous work.

If I understood Figure 3A and the Methods correctly, only in monkey N both brain regions were recorded? If this is correct, the statement that premotor and parietal regions differ in their representations is a result of a mixed within and between subjects analysis. This should be acknowledged explicitly, as it greatly reduces the statistical power of this statement, and the repeated statement about an N of 3 is misleading. CI neurons are defined based on a seemingly arbitrary criterion: exceeding a correlation threshold based on the arbitrary division of the data into 26 bins. Given that neural representations generally span a continuum, I would like to see the distribution of 'causal inference effects' for individual neurons, e.g. in form of a distribution of r2 values (obtained from the correlation; or a regression as I suggest below). The apparent difference between brain regions (Figure 3G) may simply result from the specific choice of statistical cutoff (the criterion of p<0.05 becomes meaningless in the presence of 475+238 tests in total). Seeing the individual-neuron data here seems vital.

The analysis of population timing suggests that premotor cortex leads (Figure 4F). Is it possible to extract by how much time? Also, the authors focus on the encoding of Pcom, which comprises both the a priori binding tendency and the discrepancy. Why did the authors not decode both the prior and the current multisensory discrepancy separately? This would seem important to differentiate neural signatures of priors from those of current sensory signals.

Showing the actual data: The key results (e.g. Figure 3C; 4F; 5B; 5F; 6C) would be much stronger rand more convincing if the actual units of analysis were shown in some ways. How does decoding accuracy vary across neurons?

Other details:

For most tests there are no measures of effect sizes reported, sometimes the respective test-statistics is missing, and the degrees of freedom remain very unclear. I understand that they wary between analysis, but given that some tests are based on the actually recorded units, some of pseudo-trials or binned data, it would be very important to report for each test the assumed independent units and their number. The false-discovery rate is mentioned frequently, but the precise method is not stated. Most analyses are based on Wilcoxon tests, but figures show mean and SDs. I encourage the authors to use the same nonparametric (or parametric) approaches for figures and stats (e.g. show boxplots and individual data). L. 892: what was precisely compared with the ANOVA? Cluster-based tests: the parameters and the procedures for this test are not reported (l.974)

To determine whether a neuron confirms to the expectation of causal inference, why is it necessary to bin the data (l 893ff)? Could one not simply derive a regression model for each neuron and visualize the R2 or F-value?

The authors seem to interpret differences in the significances (e.g. of cluster-based permutation tests) as significant differences between regions and as establishing differences in the relative timing of effects. These are statistical fallacies (e.g. Sassenhagen https://doi.org/10.1111/psyp.13335; and Makin https://doi.org/10.7554/*eLife*.48175).

For every statement claiming differences between parietal and premotor cortex it is necessary to directly impellent the respective contrast between neurons in each brain region to support such a difference.

Other methods:

Spike sorting: I could not find criteria used for spike sorting. Where the analyzed units single units or MUA? More details about spike thresholds, cluster separation etc. should be provided.

The total number of switches between blocks (e.g. P following VPC) should be reported, as this constitutes the effective degrees of analysis of the block switching analysis (Figure 2C).

Causal inference models and optimization: The methods leave it unclear how the two alternatives of common and separate sources were combined in the BCI model. Previous work has explored a number of decision functions (e.g. Rohe and Noppeney's work, or Wozny et al. PlosCompBiol 2010) but for the present study it remains unclear which decision function was used. Model fitting: how were likelihoods computed and the posteriors sampled for model fitting? I feel that the procedures are not described in sufficient detail to be reproduced. Over what range of disparities was the model optimized? This is important for the Null model mentioned later on. What is the precise number of data points that entered the BIC calculation?

Markov analysis: If I understood it correctly, SigmaA and SigmaV are fit to the entire block, and the Pprior derived from the entire block was used as starting value for this parameter? The authors conclude (l 256ff) that to 'maintain a consistency of causal inference, sensory uncertainty … is updated ' as well. However, the Markov model seems to focus only on the updating of the prior.

Processing of single unit data: In my view the paper would profit from showing actual single neuron PSTH's and how smoothing effected these. The methods (l. 857) mention a 400ms sliding window, but the periods of interest (e.g. target holding) are only minimally longer than this (500ms). This makes we worried that the analyzed data effectively blurs neural representations across epochs and is affected by movement artifacts. When computing the modality contributions to each response, what task epoch was analyzed to derive the tuning curves (l. 869ff)?

Overall there are many seemingly arbitrary choices in the methods. These include the thresholds to define neurons as 'causal inference', the number of trials required for neurons to be included in the population analysis (l. 964), the duration of smoothing kernels and temporal analysis windows (l. 848 ff), the binning of data for neuro-behavioral correlation (l. 893ff), in the generation of population patterns (l. 912ff), in the cluster-based test (not reported!). It would be good to see a justification for these choices or to learn whether the authors ensured that their main results do not depend on these precise choices.

4) The authors do not justify why they recorded in the transition between F4 (rostral ventral premotor) and F5 (caudal ventral premotor) with head visual/tactile optic flow signals and grasping signals respectively. The obvious target is F2 (dorsal premotor) since it has strong reaching signals and is highly connected with area 5 of the parietal lobe (Rizzolati, 1990; Mendoza and Merchant, 2014). The authors should provide a more detailed account on the areas studied and the criteria adopted to localize the recording sites. The specification that they were "determined by individual MRI atlas" does not warrant for areal identification, because on natural variability. On this regard, the Figure 3A' insets should be adjusted (for parietal recording sites in L and R hemispheres, the sulci orientation should be different, and for the premotor ones sulci should be reported).

5) For the discussion and comparison to previous work: The paradigm focuses on visuo-motor paradigm in which the sensory cues are both generated by the subject itself. In contrast, in many classical (e.g. audio-visual; or visual-vestibular)) paradigms both sensory cues are external in nature, and not linked to the subject's action. While in both types of paradigm sensory cues are integrated and can also induce perceptual recalibration, the visuo-motor paradigm still is conceptually distinct and this has implications for the interpretation of the results. The authors should discuss whether they believe that their findings generalize to other paradigms and whether the same or possibly distinct (e.g. parietal) brain regions should be investigated during such paradigms. Such a discussion seems important to place the present work in the context of the plethora of previous work. Indeed, the present study is completely lacking discussion of results with respect to current knowledge on the functional properties of premotor and parietal neurons subtending reaching. The literature on this topic is vast, but the following studies, as examples, could be relevant in this context:

1. Archambault et al. J Neurosci 2011 (comparison on premotor vs parietal, where premotor activity leads parietal one)

2. Caminiti et al. eNeuro, 2017 (overall picture of connectivity of fronto-parietal network with updated literature on functional properties of different areas)

3. Caminiti et al. J Neurosci 1991 (first paper on encoding of reaching in Premotor cortex)

4. Churchland MM, et al. Nature 2012

5. Cisek and Kalaska, Neuron 2005 (on premotor activity during reaching)

6. Gail and Andersen J Neurosci 2006 (on neural dynamics of sensorimotor transformations in parietal cortex)

7. Jerjian SJ, Sahani M, Kraskov A *ELife* 2020 (on movement representation in premotor cortex)

8. Jiang X et al. Cell Rep 2020 (onpremotor neural Activity during Observed and Executed Movements)

9. Mountcastle et al. J Neurophysiol 1975 (first pioneering study on the role of parietal cortex in visuomotor control)

10. Pezzulo et al. Progr Neurobiol 2022 (on the neural dynamics of premotor neurons during action execution and observation)

11. Santhanam et al. J Neurophysiol 2009

[Editors' note: further revisions were suggested prior to acceptance, as described below.]

Thank you for resubmitting your work entitled "Neural dynamics of causal inference in the macaque frontoparietal circuit" for further consideration by *eLife*. Your revised article has been evaluated by Tirin Moore (Senior Editor) and a Reviewing Editor.

The manuscript has been improved but there are some remaining issues that need to be addressed, as outlined below:

The authors did a good job at answering all the reviewers' comments in the rebuttal, particularly the once regarding analysis and statistically details. However, the consensus of the reviewers is that no real changes in the structure of the paper were carried out to simplify the framing of the manuscript and make it more accessible to a larger audience. In addition, all the reviewers are concerned with the lack of rational regarding the recording locations in the main text.

*Reviewer #1 (Recommendations for the authors):*

Although the authors did a good job at answering all the reviewers' comments in the rebuttal, particularly the once regarding analysis and statistically details. However, many of the framing and conceptual comments were not really incorporated in the actual reviewed manuscript. Specifically:

1) Please start the paper by giving an intuitive example of the key problem addressed in the manuscript.

2) There is no change in the introduction and the Results sections regarding a simpler and more intuitive framing of the paper (Figure 2A is the same). Again, the reader needs to go quite further into the manuscript to understand the main question and how the authors implemented the experiment.

3) The paper should refer to the classical notions of sensory motor integration in the parieto-premotor circuit in the discussion.

4) The authors did not mention why they recorded ventral premotor and a mix of area 5 and 7a in the main text.

*Reviewer #2 (Recommendations for the authors):*

In my previous review I pointed out that, although the experimental paradigm was overall well designed and the data analysis technically sophisticated, the manuscript was flawed in several aspects, particularly in relation to the way the paper was written, and the data reported and discussed.

Despite the extensive point-to-point reply, the revision of the paper remains disappointing, as no substantial changes have been made to consider the criticisms. As a matter of fact, the new version of it is essentially identical to the original one in all its sections (Abstract, Introduction, Results and Discussion). Surprisingly enough, despite the authors' attempt to reply with accuracy to the different issue raised in the reviewing process, no significant improvement of the resubmitted manuscript was achieved, as in most instances all new information was not fully integrated in the revised version.

The authors were invited to place the present work in the context of the extensive literature on the neurophysiology of the parieto-frontal network, with special attention to its role on reaching movements. In fact, the original manuscript did not adequately discuss the results within the conceptual frame offered by the knowledge accumulated over the last forty years on the dynamic properties of premotor and parietal neurons subtending arm movements. This suggestion was completely ignored, as both Discussion and Introduction remained virtually identical across versions and the authors just added a few references, among those suggested by the reviewers, in a rather superficial fashion, without any emphasis about how they were related to findings and conclusions of the present study.

Concerning point (1) the authors' action was limited to the mere insertion of new titles at the beginning of some paragraphs. In their response, it is reported that the logic of the manuscript is outlined in the unchanged Fig. 2A, which was already present in the previous version. Therefore, no significant change has been made to take into account this aspect.

Furthermore, the selected units shown in Fig. 3D to offer an example of neural activity in form of raster plots and mean firing rates (not histograms, as stated) are not indicative of clear response modulation.

The mentioned Table 1 is neither reported in the main text, nor in Supplementary Material.

When asked to evaluate the temporal difference between premotor and parietal activity, the authors just replied that "The population decoding of Pcom (Figure 4E) indicated that the premotor cortex leads the parietal by about 300 ms", but this observation refers to what already shown in the earlier version of the manuscript. Even in this case, in fact, no change was made to the analyses and to the text to take this point into consideration.

Also concerning the spike sorting technique adopted, despite the reviewer's request, no further details have been provided, relative to what was already reported in the first version of the manuscript.

Despite the explicit request (see point 4) to provide more details on which premotor area was considered in this study, the authors persist in referring loosely to "premotor" cortex, not specifying exactly which among the different premotor areas is being studied, apart from the details provided graphically in the brain figurine. Given the different functional properties and characteristics in connectivity among different premotor regions, it is inappropriate and misleading from a neurophysiological perspective to simply refer to premotor cortex. Provided that the region of recording mainly encompasses premotor area F2vr (medial to the spur of the arcuate sulcus), a small and medial part of premotor area F5, and part of F4, as evident from Fig. 3A, the main text did not report the rationale underlying the selection of the F2vr, F4/F5 for the present study. In addition, according to the new details provided in the current version of Fig. 3A on recording sites, some of the penetrations (corresponding to about 40-45 collected units) belong to prefrontal area 8 (FEF). These cells should be removed from the premotor database.

Furthermore, the lack of precise identification of the area/s of neural recording concerns parietal cortex as well. In fact, from a careful inspection of Fig. 3A, most of the penetrations in Monkey N belong to area 7 (which is part of the Inferior Parietal Lobule), and not to area 5 (which extends over the Superior Parietal Lobule instead), as stated throughout the manuscript. Finally, in all four insets of Fig. 3a it is not specified what the dashed grey lines refer to. This uncertainty would require a major revision of the parietal database.

Beyond the graphical (subjective) representation, no other information is provided about the criteria adopted to identify the recording areas and sites. First, as pointed out by the reviewers, the specification that they were "determined by individual MRI atlas" does not warrant for any areal identification; second is not even vaguely informative on how frontal and parietal areas were identified. The sentence that "The location of the recording chamber on each animal was determined by an individual MRI atlas" remained unchanged in the revised version of paper, without any further details, not even providing the reference on which Atlas has been used.

Finally, another reason of concern highlighted in the revision process refers to the lack of eye movements data, given that eye position and saccade direction exerts a well-known, although quantitatively different, influence on premotor and parietal neural activity. The authors did not refer to this critical aspect at all, nor did they discuss how eye-related signals might have influenced and eventually contaminated the reported findings.

*Reviewer #3 (Recommendations for the authors):*

The authors have addressed most reviewer comments to a sufficient degree. The work is still very dense given the large number of analyses implemented, but I have no specific suggestions for how to change this.

One remaining shortcoming is that I did not see a specific rationale for the choice of the precise recording locations in the manuscript. In reply to my previous comment, the authors have provided some rather generic text in the rebuttal, but ideally, a clear rationale for the choices should be in the manuscript.

---

## [Author Response]

Essential revisions:Main comments:The present results are novel and provide important notions. However, we have a series of important concerns.1) First, the paper is difficult to follow. The authors should use a simpler and more intuitive framing of the paper to make it more accessible to a general audience. A reader needs to integrate a large part of the results to really have an idea of what is the paper about and how the authors tested their hypothesis. Indeed, the writing in part uses words or phrases that are 'unusual' or imprecise, which makes the text difficult to understand.

In the revised manuscript, we have divided the behavior results into separate sections and summarized the conclusion of each part. In brief, the logic of the paper is outlined in Figure 2A, in which we decomposed the BCI model into sensory and hidden components, including the posterior probability of common source, the updating of prior belief, and sensory uncertainty, and then examined the behavioral evidence and corresponding neural representations respectively.

2) Another important concern is that the paper does not show the basic response properties of neurons on both areas across tasks. It is not clear how the neural activity changes between the VP, P, and VPC tasks. It is a change in preferred direction, in the width of the tuning function or on the gain modulation? We strongly suggest providing two 'layers' of information, one based on standards plotting and analysis, to be followed by a second layer of data investigation, addressed to more detailed computational aspects.

We have added the results with single-neuron analysis accordingly.

(i) We first examined whether neurons in the premotor and parietal cortex during the target-holding period were selective to basic task components, including condition (VP or P task), arm location, and visual disparity. For each neuron, we conducted a Two-ANOVA in two datasets. One dataset contains the VP and P tasks (condition (2 levels: VP and P tasks) × arm location (5 levels: [-30°, -20°, 0°, 20°, and 30°]); the response variable is the mean firing rate during the holding period of each neuron). If the main effect of condition (or arm location) in the Two-ANOVA was found (*p* < 0.05), this neuron was classified as a condition (or arm location) selective neuron. The other dataset is the VPC task (the visual disparity (9 levels: [-45°, -35°, -20°, -10°, 0°, 10°, 20°, 35°, 45°]) × target position (5 levels: [-30°, -20°, 0°, 20°, and 30°]); the response variable is the mean firing rate during the holding period of each neuron). If the main effect of visual disparity in the Two-ANOVA was found (*p* < 0.05), this neuron was classified as a disparity selective neuron. We found that in the premotor cortex, 39% (186/475) of neurons were selective to condition, 21% to arm location, and 34% to visual disparity (Figure 3—figure supplement 1A, upper panel). In the parietal cortex, 35% (83/238) of neurons were selective to Condition, 26% to arm location, and 31% to visual disparity (Figure 3—figure supplement 1 lower panel, ANOVA, main effect, *p* < 0.05). Over 25% of neurons in both regions were selective to multiple variables (premotor, 119/475; parietal, 64/238). Figure 3—figure supplement 1B represents example selective neurons in the premotor and parietal cortex, respectively.

(ii) For the neurons with arm-location selectivity (Figure 3—figure supplement 1A, premotor, n = 100; parietal, n = 63), we further investigated whether the tuning functions of these neurons were modulated in VPC (0°) task relative to the control condition (VP task). We fitted the tuning curve with a reduced von Mises function by using the neuron response in different arm locations (5 levels: [−30°, −20°, 0°, 20°, and 30°]) for these two tasks respectively. Here VPC (0°) task represents the trials in the VPC task where the disparity equals 0. And the fitting function was defined as:fr(x)=b+a∗cos(x−μ) where *b* is the spontaneous firing rate of the neuron, *a* is defined as the gain index, and μ is the preferred arm location. fr(x) represents the firing rate when the arm location is x. In the following, we analyzed the spontaneous firing rate, gain index, and preferred arm location of each neuron between the VP and VPC (0°) tasks in both premotor and parietal cortices. The results show that the preferred arm location was not significantly changed in both regions (Author response image 1 right, premotor, *p* = 0.62; parietal, *p* = 0.29, Wilcoxon signed-rank test). The spontaneous firing rates during the VP task were higher than that during the VPC (0°) task in the parietal cortex (Author response image 1 left, parietal, Wilcoxon signed-rank test, *p* = 0.012) but not in the premotor cortex (Author response image 1 left, premotor, Wilcoxon signed-rank test, *p* = 0.55). The gain index during the VP task was higher than that during the VPC (0°) task in the parietal cortex (Author response image 1 middle, parietal, Wilcoxon signed-rank test, *p* = 0.0012) but not in the premotor cortex (Author response image 1 middle, Wilcoxon signed-rank test, *p* = 0.081). The result is consistent with the conclusion that parietal, but not premotor, neurons update the neural information about arm locations between the VP and VPC (0°) tasks (Figure 6B).

**Author response image 1. sa2fig1:** The comparisons of tuning curve parameters between the VP and VPC (0°) tasks. Left: The spontaneous firing rates during VP task were higher than that during VPC (0°) task in parietal cortex (parietal, *W* = 641.0, df = 62, r_rb_ = 0.36, *p* = 0.012) but not in the premotor cortex (*W* = 2,350.0, df = 99, r_rb_ = 0.069, *p* = 0.55). Middle: The gain index during VP task were higher than that during VPC (0°) task in parietal cortex (parietal, *W* = 535.0, df = 62, r_rb_ = 0.47, *p* = 0.0012) but not premotor cortex (*W* = 2,017.0, df = 99, r_rb_ = 0.20, *p* = 0.081). Right: The preferred arm location was not significantly changed in the premotor and parietal cortex (premotor, *W* = 2,334.0, df = 99, r_rb_ = 0.056, *p* = 0.62; parietal, *W* = 852.0, df = 62, r_rb_ = 0.15, *p* = 0.29). Pair-wise comparisons were performed using Wilcoxon signed rank test. Effect sizes (r_rb_) were performed using the rank-biserial correlation.

(iii) Furthermore, we reported raster plots and histograms of two example neurons (integration- and segregation- preferred) that varied their responses to the visual disparity during the target-holding periods (gray zones) in Figure 3D.

3) Statistics:Results of model fitting: The methods mention that the authors compared three models: segregation, fusion and causal inference. However, in the results (Table S1) only two models are presented. The results for segregation should be reported for the sake of completeness. The Ppriori is high, e.g. 0.98 and 0.999. This makes me wonder in how far the R2 and the EPs' can be so much different between causal inference and fusion. Considering e.g. model averaging decision function for BIC, a prior of 0.999 would make fusion and BCI nearly identical. I feel that either the Methods section is lacking important details about the models (See above) or there is either a mistake in the analysis or the reported numbers in the table. The analysis of Bayesian models seems to lack major details, the statistical reporting is below standard (missing effect sizes, degrees of freedom, lack of individual data in figures), the study shows many unjustified parameter choices and key results seem to lack statistical support: not all statements about differences between parietal and premotor cortex seem supported by a direct statistical comparison. Further, while three monkeys contributed data, for only one does the study report data from both brain regions; this makes the claim of a difference between brain regions rather weak and this shortcoming needs to be clearly acknowledged. The actual underlying data (e.g. how single neuron responses are converted to tuning curves; how decoding accuracies vary across neurons) is not shown, which makes it difficult to interpret the robustness of the results. In particular, as the units of analyses vary tremendously between Figures (experimental blocks, neurons, pseudo-epochs, etc).

We thank the referee for the suggestions. The details of the model have been added to the revised Methods.

(i) The full segregation model fitting results are now included in Author response table 1.

**Author response table 1. sa2table1:** Model parameters and fitting evaluations of two models for monkeys.

subject	Causal inference (model averaging)	Forced fusion									
	relBIC_group_	EP	R^2^	**σ** _P_	**σ** _V_	*Prior*	relBIC_group_	EP	R^2^	**σ** _P_	**σ** _V_
Monkey H	0	1	0.96±0.0017	7.72±0.14	5.83±0.090	0.999±0.0004	329.52	0	0.93±0.0039	9.87±0.24	9.02±0.23
Monkey N	0	1	0.93±0.0080	9.56±0.16	4.93±0.15	0.86±0.026	812.34	0	0.37±0.36	11.34±0.10	10.46±0.13
Monkey S	0	1	0.96±0.0022	8.98±0.14	5.72±0.22	0.98±0.012	290.04	0	0.94±0.0027	10.10±0.14	8.34±0.17
subject	Full segregation (proprioceptive only)	Full segregation (visual only)									
	relBIC_group_	EP	R^2^	**σ** _P_	relBIC_group_	EP	R^2^	**σ** _V_			
Monkey H	1876.10	0	0.44±0.026	14.43±0.15	1677.04	0	0.58±0.019	14.51±0.091			
Monkey N	1902.82	0	0.53±0.018	14.76±0.070	1739.40	0	0.51±0.078	14.38±0.095			
Monkey S	1937.32	0	0.39±0.040	14.72±0.066	1440.47	0	0.71±0.016	12.75±0.28			

The model parameters and *R*^2^ were averaged across days for monkeys; data are presented as the means ± the standard errors. The relBIC_group_ was the summation of all days’ BIC for monkeys.

Abbreviations: **σ**_P_, the standard deviation of the proprioception likelihood; **σ**_V_, the standard deviation of the vision likelihood; *P_prior_*, the prior probability of a common source; relBIC_group_, Bayesian information criterion at the group level; EP, exceedance probability; *R*^2^, coefficient of determination.

(ii) We agree with the referee that the *P_prior_* is high in the monkey H and S, and relatively low in the monkey N. Despite the high *P_prior_* in the causal inference model, we would like to mention that there is an essential difference between these two models. In particular, the long-tailed or bimodal distribution of proprioceptive drift under large disparities can only be explained by the causal inference model (in red, Figure 1—figure supplement 2 ) but not by the forced-fusion model (unimodal distributions in green, Figure 1—figure supplement 2 ) (an example behavior session in Figure 1—figure supplement 2 ).

(iii) For the analysis of Bayesian models, the analysis of the Bayesian model was reported in revised Supplementary file 1-Table 1; see the relBIC_group_ (Bayesian information criterion at the group level) and EP (exceedance probability).

(iv) Rank-biserial correlation was used to estimate the effect size for the nonparametric test (Kerby, Comprehensive Psychology, 2014). The degrees of freedom and effect size were added to the revised figure legend.

(v) We used the premotor and parietal neurons from the same monkey (Monkey N) to decode *P_com_*. The results suggest the same tendency as the results using parietal and premotor neurons across monkeys (Figure 4—figure supplement 2 ).

(vi) To verify the robustness of the decoding results, we tested the decoding ability by sampling different numbers of neurons for *P_com_*, *P_prior_*, disparity, and arm locations, respectively (Author response image 2 and 3). The conclusion holds.

**Author response image 2. sa2fig2:** Population decoding of *P_com_* (upper panel), *P_prior_* (middle panel), and Disparity (bottom panel) with different neuron numbers in the premotor (left panel) and parietal (right panel) cortex, respectively. The horizontal dashed black line represents the chance level. The horizontal solid bars at the top represent the time of significant decoding accuracy (cluster-based permutation test, *p* < 0.05). Different color shades represent different numbers of neurons used for decoding.

**Author response image 3. sa2fig3:** Population decoding of arm locations with different neuron numbers in the premotor and parietal cortex, respectively. The horizontal dashed black line represents the chance level. The flat bar at the top represents the time bins in which the decoding accuracy for the VPC (0°) task was significantly lower than that for the VP task (cluster-based permutation test, *p* < 0.05). N: number of neurons used for decoding.

The updating of the believe about the sensory causal structure is a central component of this work. The authors present this as well established aspect of the BCI model (l. 170ff). However, most previous studies used a static model that was fit to the aggregate data of an entire experiment without taking the trial history into account (as in the cited Koerding et al. 2007 paper). AS some recent work has incorporated such trial dependencies, it would be important to acknowledge these studies and to explain the novelty of the present work (e.g. Rohe et al. NatComm 2019; Beierholm et al. eLife 2020; Badde et al. Cognition 2020 may also be relevant).

The discussion about these studies has been added and the references have been cited in the revised manuscript accordingly (L. 596).

Analysis of recalibration: The analysis of P trials after the VP or VPC task effectively looks at what is known as trial-wise multisensory recalibration (see e.g. Bruns et al. Scientific Reports 2015; Park and Kayser eLife 2019; van der Burg et al. J Neurosci 2013; Wozny et al. JNeurosci 2011; Badde et al. Cognition 2020; there is extensive literature on this both in the spatial and the temporal domain!). It seems awkward to investigate this recalibration of uniusensory judgements without alluding to previous work.

Indeed, the references suggested by the referee are relevant. We have cited them accordingly (L. 616). However, it is worth noting that, instead of the multisensory recalibration of the sensory errors [the spatial localization errors (Bruns et al., 2005; Park and Kayser, 2019; Badde et al. Cognition 2020) and temporal shifts (van der Burg et al. J Neurosci 2013)], we focused on sensory uncertainty and its updating.

If I understood Figure 3A and the Methods correctly, only in monkey N both brain regions were recorded? If this is correct, the statement that premotor and parietal regions differ in their representations is a result of a mixed within and between subjects analysis. This should be acknowledged explicitly, as it greatly reduces the statistical power of this statement, and the repeated statement about an N of 3 is misleading. CI neurons are defined based on a seemingly arbitrary criterion: exceeding a correlation threshold based on the arbitrary division of the data into 26 bins. Given that neural representations generally span a continuum, I would like to see the distribution of 'causal inference effects' for individual neurons, e.g. in form of a distribution of r2 values (obtained from the correlation; or a regression as I suggest below). The apparent difference between brain regions (Figure 3G) may simply result from the specific choice of statistical cutoff (the criterion of p<0.05 becomes meaningless in the presence of 475+238 tests in total). Seeing the individual-neuron data here seems vital.

(i) We thank the referee for reminding us of the missing information. We have acknowledged the limitations of our analysis as follows (L. 1130 in the revised manuscript):

“Note that, due to the limitations of the asynchronous recording (the premotor and parietal neurons were grouped from different individual animals, and only monkey N was recorded in both areas), further studies are required to clarify the dynamics and functional interactions between regions using a simultaneous recording.”

(ii) Here we show the histogram of the Pearson correlation coefficients between *P_com_* and VP weight (Author response image 4). The correlation coefficients of both regions are significantly larger than 0 (two-sided one-sample t-test, Premotor: *t_474_* = 7.88, *p* < 0.001, Cohen’s d = 0.36; Parietal: *t_237_* = 3.16, *p* < 0.01, Cohen’s d = 0.21). The Pearson correlation coefficients in premotor neurons are slightly higher than that in the parietal neurons (two-sided two-sample t-test, *t_474, 237_* = 1.82, Cohen’s d = 0.10, *p* = 0.069).

**Author response image 4. sa2fig4:** Histogram of Pearson correlation coefficients between VP weight and *P_com_*. The correlation coefficients of both regions are significantly larger than 0 (two-sided one-sample t-test, Premotor: t_474_ = 7.88, Cohen’s d = 0.36, *p* < 0.001; Parietal: t_237_ = 3.16, Cohen’s d = 0.21, *p* < 0.01). ** *p* < 0.01; *** *p* < 0.001.

The analysis of population timing suggests that premotor cortex leads (Figure 4F). Is it possible to extract by how much time? Also, the authors focus on the encoding of Pcom, which comprises both the a priori binding tendency and the discrepancy. Why did the authors not decode both the prior and the current multisensory discrepancy separately? This would seem important to differentiate neural signatures of priors from those of current sensory signals.

(i) The jPECC analysis only suggested a leading tendency from premotor to parietal. The population decoding of *P_com_* (Figure 4E) indicated that the premotor cortex leads the parietal by about 300 ms.

(ii) We have presented the decoding result of the *P_prior_* in Figure 5B and compared it with the *P_com_* decoding result in Figure 5F. We found that the premotor neurons encode the *P_prior_* (previous trial’s *P_com_*) during the baseline period, where there is no sensory input.

(iii) We have now reported the disparity decoding results in Figure 5—figure supplement 2, and verified it with the number of units (Author response image 5). The result of population decoding (Author response image 5) showed a systematic sequential procession of causal inference in the frontoparietal circuit again. At first, the brain encodes the *P_prior_* before the multisensory disparity onset (the *P_prior_* can be decoded at the baseline period), after that the *P_com_* is estimated by combining the *P_prior_* and multisensory disparity. Note that the *P_com_* and disparity can only be decoded after visual stimulus onset, and the disparity signal emerges earlier than the signal of *P_com_*.

**Author response image 5. sa2fig5:** Population decoding of *P_com_* (upper panel), *P_prior_* (middle panel), and Disparity (bottom panel) with different neuron numbers of premotor (left panel) and parietal (right panel). The horizontal dashed black line represents the chance level. The horizontal solid bars at the top represent the time of significant decoding accuracy (cluster-based permutation test, *p* < 0.05). Different color shades represent different numbers of neurons used for decoding.

Showing the actual data: The key results (e.g. Figure 3C; 4F; 5B; 5F; 6C) would be much stronger rand more convincing if the actual units of analysis were shown in some ways. How does decoding accuracy vary across neurons?

We have shown the actual units of analysis for each result (see Author response image 1; Figure 3C; Author response images 2, 3 and 4). Furthermore, we have tested it with different numbers of units to decode *P_com_*, *P_prior_*, disparity, and arm locations, respectively (Author response images 2 and 3). The same conclusion holds.

Other details:For most tests there are no measures of effect sizes reported, sometimes the respective test-statistics is missing, and the degrees of freedom remain very unclear. I understand that they wary between analysis, but given that some tests are based on the actually recorded units, some of pseudo-trials or binned data, it would be very important to report for each test the assumed independent units and their number. The false-discovery rate is mentioned frequently, but the precise method is not stated. Most analyses are based on Wilcoxon tests, but figures show mean and SDs. I encourage the authors to use the same nonparametric (or parametric) approaches for figures and stats (e.g. show boxplots and individual data). L. 892: what was precisely compared with the ANOVA? Cluster-based tests: the parameters and the procedures for this test are not reported (l.974)

(i) Rank-biserial correlation was used to estimate nonparametric test effect size (Kerby, Comprehensive Psychology, 2014). The degrees of freedom and effect size were added to the revised figure legend, where a nonparametric test was performed.

(ii) Benjamini-Hochberg procedure was used to correct the false discovery rate (FDR) (see revised Methods, L. 914).

(iii) We have shown the individual data and used the boxplots (see revised Figures 2B, C, and D and 3C).

(iv) For L. 892 now L. 982, two-way ANOVA was performed with factors of Condition (2 levels: VP and P) and target location (5 levels).

(v) The detail of cluster-based permutation tests was added in Methods (L. 1059) as follows:

“The significant time duration was determined using a cluster-based permutation test for multiple comparisons across time intervals (permutations = 5000; cluster-level statistic: sum of the t values in a cluster; auxiliary cluster defining threshold t = 3) (Gramfort et al., 2013).”

To determine whether a neuron confirms to the expectation of causal inference, why is it necessary to bin the data (l 893ff)? Could one not simply derive a regression model for each neuron and visualize the R2 or F-value?

Since the *P_com_* is a hidden variable predicted by the BCI model, it cannot be directly obtained from the drift of one trial. To estimate the *P_com_* of each trial, we first simulated 5000 trials under each disparity using the fitted BCI model and grouped these trials into 29 bins. Thus, the *P_com_* of each simulated trial can be obtained by the BCI model. The *P_com_* of the simulated trials within each bin were averaged to obtain the *P_com_* of each bin. Finally, each behavior trial was assigned to one of these bins according to its proprioceptive drift and disparity, and the *P_com_* of each trial was assigned according to the bin it belongs to. Thus, the correlation between VP-weight and *P_com_* can be calculated across 29 bins.

The authors seem to interpret differences in the significances (e.g. of cluster-based permutation tests) as significant differences between regions and as establishing differences in the relative timing of effects. These are statistical fallacies (e.g. Sassenhagen https://doi.org/10.1111/psyp.13335; and Makin https://doi.org/10.7554/eLife.48175).

We do not rely upon the cluster-based permutation tests to determine differences in the significance as significant differences between regions. Rather, we adopted a randomization test (Panichello and Buschman, bioRxiv, 2020) ( also see Methods, L. 1063):

“To test whether premotor neurons encode *P_com_* earlier than parietal neurons, a randomization test was performed between them. Neurons with more than 50 trials in each *P_com_* group were included in this analysis. The corresponding numbers (here, 50 neurons per region) of neurons were randomly exchanged between the paired regions 1,000 times to generate a null distribution (chance level) of time lags. The significance was determined by a permutation test of the true time lag from the original data and the null distribution (Panichello and Buschman, 2020).”

For every statement claiming differences between parietal and premotor cortex it is necessary to directly impellent the respective contrast between neurons in each brain region to support such a difference.

We have added the direct comparison results in the revised manuscript and response letter. In brief, at the single-neuron level, we directly compared the neural information for the arm location (Figures 3C, Author response image 1, and Figure 6B) and *P_com_* (Figures 3H and Author response image 4) between the parietal and premotor cortex. At the population level, we conducted population decoding (Figures 4E, 5B, and 6C) and the joint peri-event canonical correlation (jPECC) analysis (Figure 4F) to demonstrate the difference between these regions.

Other methods:Spike sorting: I could not find criteria used for spike sorting. Where the analyzed units single units or MUA? More details about spike thresholds, cluster separation etc. should be provided.

All spike data were re-sorted using off-line spike sorting clustering algorithms (Plexon, principal component analysis). With manual adjustments, only well-isolated units were considered for further analysis (signal-to-noise is more significant than 3). As shown in the revised Methods (L. 786):

“On-line raw neural signals were processed offline to obtain a single unit by Offline Sorter (Plexon Inc, Dallas, TX). All spike data were re-sorted using off-line spike sorting clustering algorithms (Plexon, principal component analysis). The auto sorted neurons were then refined manually, and only well-isolated units were considered for further analysis (signal-to-noise is larger than 3). The sorted files were then exported as MATLAB format for further analysis in MATLAB (Mathworks, Natick, MA, USA) and Python (The Python Software Foundation).”

The total number of switches between blocks (e.g. P following VPC) should be reported, as this constitutes the effective degrees of analysis of the block switching analysis (Figure 2C).

There are 282 P bocks following VPC blocks and 240 P bocks following VP blocks.

Causal inference models and optimization: The methods leave it unclear how the two alternatives of common and separate sources were combined in the BCI model. Previous work has explored a number of decision functions (e.g. Rohe and Noppeney's work, or Wozny et al. PlosCompBiol 2010) but for the present study it remains unclear which decision function was used. Model fitting: how were likelihoods computed and the posteriors sampled for model fitting? I feel that the procedures are not described in sufficient detail to be reproduced. Over what range of disparities was the model optimized? This is important for the Null model mentioned later on. What is the precise number of data points that entered the BIC calculation?

(i) We used the model average decision function to estimate the final arm location. We have revised the methods accordingly in the Methods (L. 836):

“Here, we used model average decision function to estimate final arm location (Fang et al., 2019):

S^P=p(C=1|xV,xP)S^VP,C=1+(1−p(C=1|xV,xP))S^P,C=2.”

(ii) We used maximum likelihood estimation for model fitting, which is described in the Methods (L. 844):

“To estimate the best-fitting model parameters in the BCI model, for each recording session, an optimization search was implemented that maximized the log likelihood of each model given the monkey’s data under the VPC task. The prior probability of a common source (*P_prior_*) and visual and proprioceptive standard deviations, σ*_V_* and σ*_P_*, respectively, were set as free parameters to be optimized. Five thousand trials per disparity were simulated for each optimization step to obtain the distribution, and the sum log likelihood of the observations given the model was calculated for each disparity. Then, the parameters were optimized by minimizing the sum log likelihood using a genetic algorithm (ga function in MATLAB).”

(iii) The total sessions of each monkey that entered the BIC calculation were 68 days (Monkey H), 90 days (Monkey N), and 85 days (Monkey S), respectively.

Markov analysis: If I understood it correctly, SigmaA and SigmaV are fit to the entire block, and the Pprior derived from the entire block was used as starting value for this parameter? The authors conclude (l 256ff) that to 'maintain a consistency of causal inference, sensory uncertainty … is updated ' as well. However, the Markov model seems to focus only on the updating of the prior.

(i) Yes, Σ P, Σ V, and *P_prior_* derived from the entire block were used as starting values for the parameter, and only the *P_prior_* was free in the Markov model fitting (see Methods L. 887):

“During the model fitting, we first used the Bayesian causal inference model (as mentioned before) to search the overall *P_prior_*, σ*_P_*, and σ*_V_* for each session/day, which were used as initial parameter in the subsequent Markov model. The σ*_P_* and σ_*V*_ were fixed during the model fitting.”

(ii) The referee raised an interesting question. However, as the sensory uncertainty cannot be estimated at a trial-by-trial level, the Markov model did not include it.

Processing of single unit data: In my view the paper would profit from showing actual single neuron PSTH's and how smoothing effected these. The methods (l. 857) mention a 400ms sliding window, but the periods of interest (e.g. target holding) are only minimally longer than this (500ms). This makes we worried that the analyzed data effectively blurs neural representations across epochs and is affected by movement artifacts. When computing the modality contributions to each response, what task epoch was analyzed to derive the tuning curves (l. 869ff)?

We have verified the decoding with different smoothing windows (200, 300, and 400 ms) (Author response image 6). The same conclusion holds.

**Author response image 6. sa2fig6:** Population decoding of *P_com_* with various smoothing window sizes (200, 300, and 400 ms), respectively. The horizontal dashed black line represents the chance level. The horizontal solid bars at the top represent the time of significant decoding accuracy (cluster-based permutation test, *p* < 0.05).

Overall there are many seemingly arbitrary choices in the methods. These include the thresholds to define neurons as 'causal inference', the number of trials required for neurons to be included in the population analysis (l. 964), the duration of smoothing kernels and temporal analysis windows (l. 848 ff), the binning of data for neuro-behavioral correlation (l. 893ff), in the generation of population patterns (l. 912ff), in the cluster-based test (not reported!). It would be good to see a justification for these choices or to learn whether the authors ensured that their main results do not depend on these precise choices.

We are grateful for the referee’s suggestion and have revised the Methods accordingly.

(i) The definition of ‘causal inference neuron’ follows our previous study (Fang et al., PNAS, 2019). The basic idea here is that the pattern of neuronal activity across 29 bins (see the above response) should correlate with the *P_com_* predicted by the BCI model. The significant level of *p* = 0.05 of this correlation was used as the threshold to define the ‘causal inference neuron’.

(ii) Only neurons with at least 50 trials of each condition were included in the analysis. We have added this information in the revised Methods (L. 1064).

(iii) The duration of smoothing kernels and temporal analysis windows were often used in previous studies (Fang et al., PNAS, 2019; Gu et al., Cerebral Cortex, 2016; Fried et al., Neuron, 2011).

(iv) As the neurons in our study did not record simultaneously, we generated a pseudo-simultaneous population by the bootstrap method.

(v) For the cluster-based test, the details of cluster-based tests were added in the revised Methods (L. 1059) as follows:

“The significant time duration was determined using a cluster-based permutation test for multiple comparisons across time intervals (permutations = 5000; cluster-level statistic: sum of the t values in a cluster; auxiliary cluster defining threshold t = 3) (Gramfort et al., 2013).”

4) The authors do not justify why they recorded in the transition between F4 (rostral ventral premotor) and F5 (caudal ventral premotor) with head visual/tactile optic flow signals and grasping signals respectively. The obvious target is F2 (dorsal premotor) since it has strong reaching signals and is highly connected with area 5 of the parietal lobe (Rizzolati, 1990; Mendoza and Merchant, 2014). The authors should provide a more detailed account on the areas studied and the criteria adopted to localize the recording sites. The specification that they were "determined by individual MRI atlas" does not warrant for areal identification, because on natural variability. On this regard, the Figure 3A' insets should be adjusted (for parietal recording sites in L and R hemispheres, the sulci orientation should be different, and for the premotor ones sulci should be reported).

(i) The reaching task we used in our study is an adapted task of rubber hand illusion focused on the multisensory representation of the arm. Previous studies have shown that the ventral premotor cortex and parietal area 5 are highly related to body representation and multisensory perception (see reviews (Blanke, Nat Rev Neurosci, 2012; Graziano and Botvinick, Common Mechanisms in Perception and Action, 2002)). More specifically, ventral premotor neurons have been shown to respond to visual stimuli in the space adjacent to the hand or arm (Graziano et al., Science, 1994), and parietal (area 5) neurons have been shown to respond to the seen position of a dummy arm (Graziano et al., Science, 2000). Furthermore, human fMRI studies have shown that the strength of illusion of the dummy arm was highly correlated to the neural response of ventral premotor (Ehrsson et al., Science, 2004).

(ii) We thank the referee for the suggestion and have adjusted the Figure 3A insets in the revised manuscript.

5) For the discussion and comparison to previous work: The paradigm focuses on visuo-motor paradigm in which the sensory cues are both generated by the subject itself. In contrast, in many classical (e.g. audio-visual; or visual-vestibular)) paradigms both sensory cues are external in nature, and not linked to the subject's action. While in both types of paradigm sensory cues are integrated and can also induce perceptual recalibration, the visuo-motor paradigm still is conceptually distinct and this has implications for the interpretation of the results. The authors should discuss whether they believe that their findings generalize to other paradigms and whether the same or possibly distinct (e.g. parietal) brain regions should be investigated during such paradigms. Such a discussion seems important to place the present work in the context of the plethora of previous work. Indeed, the present study is completely lacking discussion of results with respect to current knowledge on the functional properties of premotor and parietal neurons subtending reaching. The literature on this topic is vast, but the following studies, as examples, could be relevant in this context:1. Archambault et al. J Neurosci 2011 (comparison on premotor vs parietal, where premotor activity leads parietal one)2. Caminiti et al. eNeuro, 2017 (overall picture of connectivity of fronto-parietal network with updated literature on functional properties of different areas)3. Caminiti et al. J Neurosci 1991 (first paper on encoding of reaching in Premotor cortex)4. Churchland MM, et al. Nature 20125. Cisek and Kalaska, Neuron 2005 (on premotor activity during reaching)6. Gail and Andersen J Neurosci 2006 (on neural dynamics of sensorimotor transformations in parietal cortex)7. Jerjian SJ, Sahani M, Kraskov A ELife 2020 (on movement representation in premotor cortex)8. Jiang X et al. Cell Rep 2020 (onpremotor neural Activity during Observed and Executed Movements)9. Mountcastle et al. J Neurophysiol 1975 (first pioneering study on the role of parietal cortex in visuomotor control)10. Pezzulo et al. Progr Neurobiol 2022 (on the neural dynamics of premotor neurons during action execution and observation)11. Santhanam et al. J Neurophysiol 2009

We are grateful for the references that the referee suggested. The references now have been cited accordingly (L.596, 638).

Although a reaching task was used to require monkeys to report their arm locations, the neural signal about *P_com_* was observed before reaching. The current study focused on multisensory integration and causal inference, which merely relied on sensory inputs (V, P, and disparity). The reaching behavior in the task could be the action signal guided by the *P_com_*.

[Editors' note: further revisions were suggested prior to acceptance, as described below.]

The authors did a good job at answering all the reviewers' comments in the rebuttal, particularly the once regarding analysis and statistically details. However, the consensus of the reviewers is that no real changes in the structure of the paper were carried out to simplify the framing of the manuscript and make it more accessible to a larger audience. In addition, all the reviewers are concerned with the lack of rational regarding the recording locations in the main text.Reviewer #1 (Recommendations for the authors):Although the authors did a good job at answering all the reviewers' comments in the rebuttal, particularly the once regarding analysis and statistically details. However, many of the framing and conceptual comments were not really incorporated in the actual reviewed manuscript. Specifically:1) Please start the paper by giving an intuitive example of the key problem addressed in the manuscript.

We have added an intuitive example in the introduction section to highlight the main questions we addressed in this study in the revised manuscript (L. 54-58).

2) There is no change in the introduction and the Results sections regarding a simpler and more intuitive framing of the paper (Figure 2A is the same). Again, the reader needs to go quite further into the manuscript to understand the main question and how the authors implemented the experiment.

As suggested by the referee, we have revised the results and added new paragraphs of the hypotheses in the present study to give a simpler framing of the paper (Line 123-144), which explicitly elaborates Figures 1C-F.

3) The paper should refer to the classical notions of sensory motor integration in the parieto-premotor circuit in the discussion.

As suggested by the referee, we have now added the discussion about the relationship between our study and previous findings of sensorimotor representations in the frontoparietal circuit as follows (L. 682-714; L. 737-745).

“The frontoparietal circuit, including the premotor and parietal cortices, has long been recognized as a central area in sensorimotor representations (Caminiti et al., 2017; Caminiti et al., 1991). Although the present experiments shared many movement features in the reaching task, the key findings of causal inference processing are unlikely to be explained by the kinematical components. First, previous studies have demonstrated that the neuronal activities in the premotor cortex are related to hand kinematics (e.g., hand position, speed, and direction) in the motor planning and execution (Caminiti et al., 1991; Churchland et al., 2006), which lead the neural activities in the parietal cortex (Archambault et al., 2011). However, in our study, the early activities of *P_com_* in the premotor cortex cannot be purely induced by the sequential activities of kinematics in the premotor and parietal cortices. Because the *P_com_* is abstract information, and its activity pattern is not correlated with any kinematical components. Expressly, under a given value of *P_com_*, the reaching kinematics can be varied (e.g., the hand position can be anywhere on the table according to the target position and disparity in a given trial). Moreover, the neural signals about *P_com_* in the premotor cortex were observed before the target onset, where no motor planning was possible during this period. Thus, our results are consistent with the idea that the high-level information, such as abstract and hidden structures, potential probability of multiple motor options, and visual-proprioceptive integration, are encoded in the frontoparietal circuit, which could later integrate with the low-level sensory representations to guide the desired movement (Cisek and Kalaska, 2005; Gail and Andersen, 2006; Limanowski and Blankenburg, 2016).

Second, the dynamic updating of prior and sensory representation proposed a putative mechanism for multisensory recalibration in sensorimotor tasks. At the behavioral level, our results are in accord with the observations that sensory perception is modulated by a multisensory context with sensory conflicts. The BCI theory thus provides a framework to explain how the multisensory context (e.g., the prior of common source) modulates the sensory representations, such as sensory uncertainty in our study and sensory estimation (e.g., spatial localizations) in previous sensorimotor studies (Badde et al., 2020; Bruns and Roder, 2015; Park and Kayser, 2019; Van der Burg et al., 2013). The results support the notion of dynamic representations of *P_com_* in the present study – the top-down signal of common source from the premotor cortex modulates the spatial tuning in the parietal cortex and then guides hand estimation.

[…]

Intriguingly, our results seem complementary to previous findings of mirror neuron systems in the premotor and parietal cortices in both humans and monkeys. Typically, a mirror neuron fires both when individual acts and when the individual observes the same action performed by another. That is, the mirror neuron is believed to mediate the understanding of others' behavior (Jerjian et al., 2020; Jiang et al., 2020; Pezzulo et al., 2022). By contrast, the role of causal inference neurons in our study was putatively participating in self-identification and self-other discrimination. Future studies are needed to examine how these two systems work together to identify both self and foreign agents”.

4) The authors did not mention why they recorded ventral premotor and a mix of area 5 and 7a in the main text.

The recording regions were chosen based on previous monkey electrophysiological and human studies. In brief, previous studies showed that the premotor (including dorsal and ventral premotor) and posterior parietal cortices (including areas 5 and 7) are related to visual-somatosensory integration and arm recognition (see the reference below). However, there is no direct evidence showing a specific region for the visual-proprioceptive integration and causal inference processes. As the first study of causal inference integration of visual and proprioceptive signals, we decided to record neurons in both dorsal and ventral premotor and posterior cortices, including areas 5 and 7. In fact, we did not find a significant difference between the sub-regions in the premotor cortex and the sub-regions in the parietal cortex (see Author response image 7, *p* > 0.05). We have added the reasons why we chose the recorded regions in the revised manuscript as follows (see L. 296-312 in the main text).

“Previous studies showed that the premotor and parietal cortices were highly involved in body representation and multisensory perception (see reviews (Blanke, 2012; Graziano and Botvinick, 2002)). In monkeys, bimodal neurons with visual and somatosensory receptive fields were found in both premotor (including F2vr in dorsal premotor and F4/F5 in ventral premotor) and posterior parietal cortices (including area 5 and area 7) (Fogassi et al., 1999; Graziano et al., 2000; Graziano and Gross, 1993, 1998; Graziano et al., 1994). Specifically, ventral premotor neurons responded to visual stimuli in the space adjacent to the arm (Graziano and Gross, 1998; Graziano et al., 1994). The bimodal neurons in the parietal cortex (area 5 and area 7) showed to respond to both the real arm position and the seen position of a dummy arm (Graziano et al., 2000), which have a significant projection of the premotor cortex (Graziano and Gross, 1998). Consistently, human fMRI studies found that the posterior parietal and premotor (dorsal and ventral) cortices selectively respond to visual stimulation near the hand (Brozzoli et al., 2011) or the dummy hand near one’s corresponding hand (Blanke et al., 2015; Ehrsson et al., 2004). A human MEG study also revealed that the activities in the prefrontal and intraparietal sulcus were related to the causal inference computation in visual-auditory integration (Cao et al., 2019; Rohe et al., 2019)”.

**Author response image 7. sa2fig7:** (A) Left and middle: the fraction of selective neurons in the dorsal premotor (PMd) cortex and the ventral premotor (PMv) cortex (ANOVA, main effect, *p* < 0.05). There was no significant difference between the fraction of selective neurons in PMd and PMv (Pearson's chi-square test, χ^2^ = 0.013, df = 2, *p* = 0.99). Right, the fraction of causal inference neurons in the dorsal premotor cortex and ventral premotor cortex. There was no significant difference between the fraction of causal inference neurons in PMd and PMv (χ^2^ = 0.15, df = 1, *p* = 0.70). (B) Left and middle: the fraction of selective neurons in the parietal Area 5 cortex and Area 7 cortices (ANOVA, main effect, *p* < 0.05). There was no significant difference between the fraction of selective neurons in parietal Area 5 and Area 7 cortices (Pearson's chi-square test, χ^2^ = 0.047, df = 2, *p* = 0.98). Right, the fraction of causal inference neurons in the parietal Area 5 cortex and the parietal Area 7 cortex. There was no significant difference between the fraction of causal inference neurons in parietal Area 5 and Area 7 (χ^2^ = 1.023×10^-30^, df = 1, *p* = 1).

Reviewer #2 (Recommendations for the authors):In my previous review I pointed out that, although the experimental paradigm was overall well designed and the data analysis technically sophisticated, the manuscript was flawed in several aspects, particularly in relation to the way the paper was written, and the data reported and discussed.Despite the extensive point-to-point reply, the revision of the paper remains disappointing, as no substantial changes have been made to consider the criticisms. As a matter of fact, the new version of it is essentially identical to the original one in all its sections (Abstract, Introduction, Results and Discussion). Surprisingly enough, despite the authors' attempt to reply with accuracy to the different issue raised in the reviewing process, no significant improvement of the resubmitted manuscript was achieved, as in most instances all new information was not fully integrated in the revised version.The authors were invited to place the present work in the context of the extensive literature on the neurophysiology of the parieto-frontal network, with special attention to its role on reaching movements. In fact, the original manuscript did not adequately discuss the results within the conceptual frame offered by the knowledge accumulated over the last forty years on the dynamic properties of premotor and parietal neurons subtending arm movements. This suggestion was completely ignored, as both Discussion and Introduction remained virtually identical across versions and the authors just added a few references, among those suggested by the reviewers, in a rather superficial fashion, without any emphasis about how they were related to findings and conclusions of the present study.

We thank the referee for reminding us of the changes to the Discussion and Introduction. In the revised manuscript, we first added the hypotheses at the beginning of the results to give a simpler framing of the study, explaining the logic before we showed the results. We then added the reasons we chose to record the ventral and dorsal premotor and parietal (areas 5 and 7) cortices in the results before we showed the neural data. Finally, we modified the discussion to discuss that although our experiments shared many movement features in the reaching task, the key findings of causal inference are unlikely to be explained by the kinematical components during the arm movements. In the following, we list the responses point-by-point:

Concerning point (1) the authors' action was limited to the mere insertion of new titles at the beginning of some paragraphs. In their response, it is reported that the logic of the manuscript is outlined in the unchanged Fig. 2A, which was already present in the previous version. Therefore, no significant change has been made to take into account this aspect.

1) We have added an intuitive example in the introduction section to highlight the main questions we addressed in this study as follows (L. 54-58).

“For instance, in the ventriloquism illusion, when the audience is presented with a synchronous but spatially discrepant audiovisual stimulus (e.g., a speech sound from the speaker and a visibly moving mouth of the puppet), they usually infer these audiovisual stimuli are coming from a common source and illusive perceive the speech coming from the puppet.”

2) We have added an overview of the main predictions of this study at the beginning of the results section--“Behavioral paradigm and hierarchical Bayesian causal inference model” in the revised manuscript (L.123-144).

Furthermore, the selected units shown in Fig. 3D to offer an example of neural activity in form of raster plots and mean firing rates (not histograms, as stated) are not indicative of clear response modulation.

To better illustrate the response modulation by the disparity, we plotted the mean responses of the example neuron in Figure 3D during the target-holding period (Figure 3 and Figure 3-Figure supplement 2). Figure 3 shows a “segregation (P) neuron”, which is more active during the P task and exhibited increased activity under the large disparities in the VPC task. By contrast, Figure 3 shows an “integration (VP) neuron”, which is more active during the VP task and exhibited increased activity under the small disparities in the VPC task.

The mentioned Table 1 is neither reported in the main text, nor in Supplementary Material.

We indeed reported Table 1 in our last revision, but it was required to be submitted as an independent document. In this round of revision, we resubmit it as Supplementary file 1. Model parameters and fitting evaluations of two models for monkeys.docx.

When asked to evaluate the temporal difference between premotor and parietal activity, the authors just replied that "The population decoding of Pcom (Figure 4E) indicated that the premotor cortex leads the parietal by about 300 ms", but this observation refers to what already shown in the earlier version of the manuscript. Even in this case, in fact, no change was made to the analyses and to the text to take this point into consideration.

To further confirm the significance of the temporal difference between the premotor and parietal cortices, a randomization test was performed on the population decoding of *P_com_* between these two regions. Neurons with more than 50 trials in each *P_com_* group were included in this analysis. The corresponding numbers (200 neurons per region) of neurons were randomly exchanged between the paired regions 1,000 times to generate a null distribution (chance level) of time lags, and the significance was determined by a permutation test of the true time lag from the original data and the null distribution (Panichello & Buschman, 2021). The results again showed that the *P_com_* information appeared to occur significantly earlier in the premotor cortex than in the parietal cortex (Figure 4-Figure supplement 3, randomization test, *p* < 0.01). Furthermore, we also confirmed the result using the jPECC analysis in the earlier version of the manuscript (Figure 4F) and discussed the limitation of the asynchronous single-unit recording (L. 1293-1296).

Also concerning the spike sorting technique adopted, despite the reviewer's request, no further details have been provided, relative to what was already reported in the first version of the manuscript.

We have added the details of spike sorting and the reference in the revised Methods (L. 903-910), as follows:

“All isolated neurons were recorded regardless of their activity during the task, with the recording locations varying from session to session. At each location, the raw extracellular membrane potential was sampled at 40 kHz. Online raw neural signals were processed offline to obtain a single unit by Offline Sorter (Plexon Inc, Dallas, TX). All spike data were re-sorted using offline spike sorting clustering algorithms (Offline Sorter, principal component analysis) (Merchant et al., 2013). With manual adjustments, only well-isolated units were considered for further analysis (signal-to-noise is larger than 3).”

Despite the explicit request (see point 4) to provide more details on which premotor area was considered in this study, the authors persist in referring loosely to "premotor" cortex, not specifying exactly which among the different premotor areas is being studied, apart from the details provided graphically in the brain figurine. Given the different functional properties and characteristics in connectivity among different premotor regions, it is inappropriate and misleading from a neurophysiological perspective to simply refer to premotor cortex. Provided that the region of recording mainly encompasses premotor area F2vr (medial to the spur of the arcuate sulcus), a small and medial part of premotor area F5, and part of F4, as evident from Fig. 3A, the main text did not report the rationale underlying the selection of the F2vr, F4/F5 for the present study. In addition, according to the new details provided in the current version of Fig. 3A on recording sites, some of the penetrations (corresponding to about 40-45 collected units) belong to prefrontal area 8 (FEF). These cells should be removed from the premotor database.

We thank the referee for the acute observation. We have now added the reasons why we chose the recorded regions in the revised manuscript (L. 296-312). In the revised results, we removed the FEF neurons from the neuron database, conducted the same analysis, and updated all the relevant figures accordingly (Figures 3A, 3C, 3G, 3H, 4C, 4D, 4E, 4F, 5A, 5B, 5C, 5E, 5F, 6B, 6C, Figure 3-Figure supplement 1, Figure 4-Figure supplement 1, Figure 4-Figure supplement 2, Figure 5-Figure supplement 1, and Figure 5-Figure supplement 2). Note that the same conclusion holds.

The recording regions were chosen based on previous monkey electrophysiological and human studies. **Please also see our response to Reviewer 1 (Question 5)**. The recording areas include (1) both ventral and dorsal premotor cortices; (2) both inferior (area 7) and superior (area 5) parietal lobes. Indeed, the previous studies in the parietal cortex also showed that area 7 neurons also responded to the multimodal sensory inputs. For example, Graziano and colleagues showed the visual receptive fields of the visual-somatosensory bimodal neurons in area 7 were fixed to the relevant body part, which was similar to the bimodal neurons in area 5 and premotor cortices (Graziano and Gross, 1993; Graziano et al., 2000). Our further analysis confirmed that there was no significant difference in the percentage of causal inference neurons between the subregions in the premotor and parietal cortices.

We thank the referee’s suggestions. In the revised manuscript, we have now claimed that the recording regions included both areas 5 and 7 in the parietal cortex (L. 312-315). In Figure 3A, the straight dash grey line roughly separated the dorsal and ventral parts of the premotor cortex in the middle panel. The straight dash grey line roughly indicates the middle between IPS and CS. The circular dash lines indicate the recording chambers. We have added this information in the revised manuscript (L. 338-340).

Beyond the graphical (subjective) representation, no other information is provided about the criteria adopted to identify the recording areas and sites. First, as pointed out by the reviewers, the specification that they were "determined by individual MRI atlas" does not warrant for any areal identification; second is not even vaguely informative on how frontal and parietal areas were identified. The sentence that "The location of the recording chamber on each animal was determined by an individual MRI atlas" remained unchanged in the revised version of paper, without any further details, not even providing the reference on which Atlas has been used.

We collected the structural magnetic resonance images (MRI) of three monkeys (3T, Center for Excellence in Brain Science and Intelligence Technology, Institute of Neuroscience, Chinese Academy of Sciences), while they were in an MRI-compatible Horsley-Clarke stereotaxic apparatus. The location of the recording chamber on each animal was determined by the atlas with the origin at the Ear Bar Zero (Saleem & Logothetis, 2012). The centers of implanting recording chambers were [right: 20.0 mm; forward: 10.0 mm] for the premotor cortex in Monkey N, [left: 21.9 mm; forward: 24.9 mm] for the premotor cortex in Monkey H, [right: 14.7 mm; forward: 1.1 mm] for the parietal cortex in Monkey N, and, [left: 17.0 mm; forward: 3.5 mm] for the parietal cortex in Monkey S (see Author response image 8). We have added this information in the revised manuscript (L. 891-901).

**Author response image 8. sa2fig8:** The centers of implanting recording chambers in MRIs for individual monkeys. CS, central sulcus; IPS, intraparietal sulcus. Note that the recording locations of monkey N’s premotor and parietal cortices are from different structural magnetic resonance images.

Finally, another reason of concern highlighted in the revision process refers to the lack of eye movements data, given that eye position and saccade direction exerts a well-known, although quantitatively different, influence on premotor and parietal neural activity. The authors did not refer to this critical aspect at all, nor did they discuss how eye-related signals might have influenced and eventually contaminated the reported findings.

We trained the monkeys to perform the task without their eye fixed, but the eye movement during the recording sessions was recorded. To examine whether the updating of sensory uncertainty was correlated with the uncertainty of eye position between VP and VPC (0°) tasks. We identified the eye fixation position at the target holding period. We examined the divergence of eye fixation position in VP and VPC (0°) tasks and found that there was no significant difference between these tasks (Figure 2-Figure supplement 2, Wilcoxon signed-rank test, *p* = 0.81).

Moreover, to examine whether the neural activity of *P_com_* was correlated with the eye position, we calculated the Pearson correlation coefficients between eye fixation position and VP weight. We found that there was no correlation between the VP weight and the eye fixation position at the population level for both regions (the premotor and parietal cortices) at both horizontal and vertical directions (Figure 3-Figure supplement 3, Wilcoxon signed-rank test, Premotor (horizontal): *p* = 0.11; Premotor (vertical): *p* = 0.86; Parietal (horizontal): *p* = 0.35; Parietal (vertical): *p* = 0.87). Note that the recorded eye movement data used in this analysis included 78 sessions for the premotor cortex and 45 sessions for the parietal cortex. And we added this information in the revised manuscript (L. 369-270, L. 410-415, L. 601-603, and L. 1032-1061).

Reviewer #3 (Recommendations for the authors):The authors have addressed most reviewer comments to a sufficient degree. The work is still very dense given the large number of analyses implemented, but I have no specific suggestions for how to change this.One remaining shortcoming is that I did not see a specific rationale for the choice of the precise recording locations in the manuscript. In reply to my previous comment, the authors have provided some rather generic text in the rebuttal, but ideally, a clear rationale for the choices should be in the manuscript.

We have added the reasons why we chose the recorded regions in the revised manuscript (L. 296-312). Please also see our response to Reviewer 1 (Question 5).